# Deep Learning with Plausible Deniability

## Abstract

Deep learning models are vulnerable to privacy attacks due to their tendency to memorize individual training set examples. Theoretically-sound defenses such as differential privacy can defend against this threat, but model performance often suffers. Empirical defenses may thwart existing attacks while maintaining model performance but do not offer any robust theoretical guarantees.

In this paper, we explore a new strategy based on the concept of plausible deniability. We introduce a training algorithm called **P**lausibly **D**eniable **S**tochastic **G**radient **D**escent (PD-SGD), which aims to provide both strong privacy protection with theoretical justification and maintain high performance. The core of this approach is a rejection sampling technique, which probabilistically prevents updating model parameters whenever a mini-batch cannot be plausibly denied. This ensures that no individual example has a disproportionate influence on the model parameters. We provide a set of theoretical results showing that PD-SGD effectively mitigates privacy leakage from individual data points. Experiments also demonstrate that PD-SGD offers a favorable trade-off between privacy and utility compared to differential privacy (i.e., DP-SGD) and empirical defense methods.

## 1 Introduction

Deep learning models (LeCun et al., 2015) have become integral components of many contemporary technological applications, ranging from image (Obaid et al., 2020) and speech recognition (Zhang et al., 2018) to natural language processing (Deng & Liu, 2018). Their ability to uncover complex patterns in data and provide high predictive accuracy has driven broad acceptance and deployment across multiple industries. However, the pervasive usage of deep learning raises significant security and privacy issues. Privacy attacks, such as membership inference attacks (Shokri et al., 2017; Ye et al., 2022; Carlini et al., 2022), have been shown to exploit vulnerabilities, compromising the confidentiality of the model's training data.

Protecting privacy while maintaining model performance is a major challenge. Current defense strategies are such that practitioners have to choose between strong privacy guarantees and high model utility. Approaches based on differential privacy (DP) (Dwork, 2006) offer strong mathematical privacy guarantees. When applied to machine learning, these approaches usually consist of clipping and adding large amounts of noise to the gradients (Abadi et al., 2016) during training, but this often results in drastic degradation of model performance. On the flip side, empirical defense strategies such as Adversarial Regularization (Nasr et al., 2018), SELENA (Tang et al., 2022) often preserve performance but come without mathematical justification that privacy is protected and thus may ultimately prove to be highly vulnerable to future (yet-to-be-discovered) attacks.

In response to these challenges, this work aims to bridge the gap between robust theoretical privacy guarantees and practical performance. We introduce a novel training algorithm called Plausibly Deniable Stochastic Gradient Descent (PD-SGD), which takes inspiration from the principle of *plausible deniability* (Bindschaedler et al., 2017). Unlike existing approaches, PD-SGD seeks to offer a novel method for private learning without compromising performance.

The innovation at the core of the proposed learning algorithm is an efficient *privacy test*, which inspects potential gradients from mini-batches before they are used to update the model parameters. This privacy test enforces that anomalous gradients — those that are not plausibly deniable — will be discarded, thereby eliminating the leakage that may otherwise result from such updates.

The paper first discusses the theoretical foundations of the PD-SGD approach, including its design and the privacy guarantees it offers. The proposed approach is then evaluated experimentally, comparing its performance and trade-offs with those of existing methods such as DP-SGD and empirical defenses. Results demonstrate that PD-SGD offers a superior privacy-utility trade-off compared to alternatives.

## 2 BACKGROUND & RELATED WORKS

### 2.1 DEEP LEARNING

In this paper, we consider supervised models to predict a target label/value in a set $Y$ for an example given features in a set $X$. The model is a function $f : X \to Y$ that is parameterized by a vector $\theta$ of trainable parameters. The model is trained using a dataset $D$ of $n$ data points $(x_i, y_i)$, $i \in [1, n]$ where $x_i \in X$ and $y_i \in Y$ and solving for the vector $\theta$ that minimizes the loss function $\mathcal{L}(\cdot)$ on $D$.

To (approximately) solve this optimization problem, we can use Stochastic Gradient Descent (SGD) (Gower et al., 2019) or one of its many variants (Haji & Abdulazeez, 2021). We focus on mini-batch SGD which we refer to as (vanilla) SGD. In each iteration, the algorithm partitions the training set into (roughly) equal-sized mini-batches, randomly picks a mini-batch, and updates the parameters according to the mini-batch's gradient. Specifically given a mini batch $B_j$, we let $g_j = \nabla_\theta \mathcal{L}(\theta, B_j) \in \mathbb{R}^k$ denote the gradient of the loss on $B_j$ with respect to the model parameters $\theta \in \mathbb{R}^k$. The update at step $t$ is therefore: $\theta_t = \theta_{t-1} - \eta g_j$, where $\eta$ is the chosen learning rate.

### 2.2 MEMBERSHIP INFERENCE ATTACKS

Membership inference attacks (MIAs) have been extensively studied in recent years (Shokri et al., 2017; Salem et al., 2018; Yeom et al., 2018; Sablayrolles et al., 2019; Long et al., 2020; Choquette-Choo et al., 2021; Carlini et al., 2022; Ye et al., 2022; Matsumoto et al., 2023; Bertran et al., 2023; Zarifzadeh et al., 2024). These are privacy attacks where the adversary aims to determine if a specific example was included in a target model's training set. Specifically, given a specific target example $(x, y)$, the adversary seeks to discern between two competing hypotheses:

- $H_0$ ("non-member" or "out"): $(x, y) \notin D$, or
- $H_1$ ("member" or "in" or '): $(x, y) \in D$.

Membership inference attacks were first introduced by Shokri et al. (2017), employing shadow models trained on data similar to the target's to emulate its behavior and generate attack data. Recent works like Ye et al. (2022) propose different attack variants aim to reduce adversarial uncertainty to improve attack effectiveness. Carlini et al. (2022) propose a Likelihood Ratio Attack while advocating focusing on increasing true positive rates at low false positive rates.

### 2.3 DEFENSES

Table 1: **Comparison between defense methods:** We compare our proposed PD-SGD with other defense methods from privacy and utility.

| Method | Privacy | Utility |
|---|---|---|
| AdvReg (Nasr et al., 2018) | Empirical | High |
| SELENA (Tang et al., 2022) | Empirical | High |
| DP-SGD (Abadi et al., 2016) | Provable | Low |
| **PD-SGD**(Ours) | **Provable** | **High** |

There exist numerous defenses against privacy attacks in general and membership inference attacks in particular. Some of these defenses provide provable guarantees, whereas others only provide empirical mitigation.

**Defenses with a provable guarantee.** Some defenses provide a formal privacy guarantee. This is the case for the most widely-used technique called Differentially Private Stochastic Gradient Descent (DP-SGD— Abadi et al. (2016)), which provably satisfies differential privacy (Dwork et al., 2006).

DP-SGD updates the model parameters iteratively like SGD, except that it bounds privacy leakage through (1) per-example clipping and (2) noise addition. Each mini-batch gradient is computed as the average over the batch's per-example gradients, but the *per-example gradients* are first clipped to have bounded $l_2$-norm. This ensures that each example has a bounded influence on the mini-batch gradient that decreases with the size of the mini-batch. Further, the mini-batch gradient is noised with isotropic Gaussian noise before being used to update the parameters.

Given a clipping threshold $C > 0$, the noisy gradient is:

$$\bar{g}_j = \frac{1}{L} \sum_i g_{j,i} \cdot \min(1, \frac{C}{||g_{j,i}||}) + \mathcal{N}(0, \sigma^2 C^2 I) \,,$$

where $L$ is the number of examples in the mini-batch, $g_{j,i}$ is the gradient vector of example $i$ in batch $B_j$, and $\sigma$ is the noise level.

Models trained this way achieve $(\varepsilon, \delta)$-*differential privacy*, where $\varepsilon > 0$ is the privacy budget. However, models' prediction accuracy often suffers significantly due to the impact of the noise (Dörmann et al., 2021) and gradient clipping (Chen et al. (2020); Qian et al. (2021); Koloskova et al. (2023)). Careful tuning of hyperparameters, and (or) use of techniques such as data augmentation (De et al., 2022) is critical to obtain the favorable utility, especially when the amount of training (or fine-tuning) data is limited (Tobaben et al., 2023). Another drawback is increased training time, and larger memory requirements, although recent research attempts to mitigate these issues (Bu et al. (2022); Beltran et al.).

**Empirical defenses.** To address the problem of low utility while still effectively thwarting membership inference, several empirical defense mechanisms have been proposed. These include Adversarial Regularization (AdvReg) (Nasr et al., 2018), SELENA (Tang et al., 2022), and so on. We select AdvReg and SELENA because they are well-known and widely used as baselines (Tang et al., 2022; Aerni et al., 2024). These defense mechanisms are applied at training time like DP-SGD.[1]

These approaches typically employ strategies such as regularization to lower the attack score, or applying knowledge distillation to mitigate the attacks. While these empirical defense mechanisms can preserve the model utility and offer some level of privacy protection, they lack provable theoretical guarantees. Consequently, it is unclear to what extent they truly eliminate sensitive information leakage or the degree to which they will be effective against future attacks, especially adaptive attacks.

To the best of our knowledge, no existing defense mechanism simultaneously offers a provable theoretical guarantee and maintains good model utility. Our proposed method, PD-SGD, is designed to help bridge this gap (see Table 1).

## 2.4 PLAUSIBLE DENIABILITY

It is often said that differential privacy provides plausible deniability. This makes sense on the basis that differential privacy ensures that the probabilities of any output on neighboring datasets (datasets that differ in exactly one example) are tightly bounded in terms of the privacy budget $\varepsilon$.

Plausible deniability as a formal privacy notion was proposed by Bindschaedler et al. (2017) in the context of synthesizing tabular microdata. In their setting, they repeatedly select a single row of a database as a "seed" and use it to probabilistically produce a new synthetic row similar to it. The problem is that this procedure may not preserve privacy since the process statistically ties the synthetic to the seed. To get around this issue, the authors formalize the notion of plausibility deniability.

Informally, a synthetic is *plausibly deniable* if we can find that in the original database, more than $T$ (integer parameter) alternative rows could have led to generating the synthetic with similar probability. This similarity in probability is determined by a ratio bounded by some $\alpha > 1$, assuming those rows have been (as a counterfactual) selected as seed. To enforce this constraint, a privacy test using rejection sampling is defined. The test ensures that if a synthetic is ever produced that does not meet the plausibility deniability constraint, it will be thrown away. With some additional randomization of this test, this procedure can be made to yield $(\varepsilon, \delta)$-differential privacy.

---

[1]There are inference time defenses such as MemGuard (Jia et al., 2019). We do not consider them, since we propose a training time defense.

# 3 PLAUSIBLE DENIABILITY FOR DEEP LEARNING

We propose a new formulation of plausible deniability that can be applied to SGD training at the level of mini-batches. To enforce plausible deniability, we implement a privacy test on the potential gradient updates from a mini-batch. If a mini-batch includes one or more examples that yield an implausible gradient (with respect to other mini-batch's gradients), we *reject* this gradient — we do not use it to update the model parameters.

## 3.1 THE ANATOMY OF PRIVACY LEAKAGE

A root cause of privacy leakage in deep learning — the kind that membership inference attacks exploit — is the disproportionate impact of including a single example in the dataset onto the model. For instance, imagine an iteration of SGD where we have selected a batch $B$ and computed its gradient vector $g$. We can consider the counterfactual of having selected a batch $B' = B \cup \{(x, y)\}$ that includes some example $(x, y)$. The crucial observation is that the gradient vector $g'$ for $B'$ may be completely different than $g$, even if the batch $B$ is large. For instance, $g'$ may point in the opposite direction, i.e., $g' = -g$, or $g' \perp g$, or even $g' = 0$. There is no guarantee that adding any example to any batch will not arbitrarily distort the gradient. The consequence for data privacy is that if the adversary observes this, directly or indirectly (through the model parameters), then they can infer membership of $(x, y)$.

DP-SGD avoids this problem by using per-example gradient clipping. In this work, we take a different approach. Instead of trying to constrain the change in the gradient that would result from adding/removing any example, we seek to detect those batches with gradients that are not plausibly deniable. We can think of such batches are "anomalous" compared to other batches, and we can simply discard any potential parameter updates based on them.

## 3.2 PRIVACY NOTION AND PRIVACY TEST

We propose to update the model parameters only if the gradient $g_i$ is *plausibly deniable*, i.e., if it is not too dissimilar to the gradients of some other mini-batches. To formalize this, we first need to add isotropic Gaussian noise to the gradient vector $g$ as $\tilde{g} = g + Z$, where $Z \sim \mathcal{N}(0, \sigma^2 I)$. Note that adding noise to the gradient in SGD is a well-known technique that has benefits for convergence (Neelakantan et al., 2015; Ziyin et al., 2022). In our case, this allows us to view each (noisy) mini-batch gradient $\tilde{g}$ as a random variable. Given this, we can define the probability that a given fixed gradient vector $\tilde{g}$ is plausibly obtained from any mini-batch gradient $g_i$, and from there the concept of a plausibly deniable gradient update.

**Definition 1.** *Let $B_1, \ldots, B_m$ be disjoint mini-batches and $g_1, \ldots, g_m$ be their associated gradient. Let $B_s$ be the chosen "seed" batch with associated gradient $g_s$. We say that batch $B_s$ is $(\alpha, \sigma, T)$-plausibly deniable if there are at least $T > 1$ distinct batches $B_i$ with $i \in [1, m]$ that satisfy:*

$$\alpha^{-1} \leq \frac{p(\tilde{g}_s - g_s)}{p(\tilde{g}_s - g_i)} \leq \alpha \,, \tag{1}$$

*where $\tilde{g}_s = g_s + Z$ for $Z \sim \mathcal{N}(0, \sigma^2 I)$. Here $\sigma > 0$, $\alpha \geq 1$, $T > 1$ are privacy parameters.*

Now let $\alpha = \exp(\gamma)$ for some $\gamma > 0$ and $p(\cdot)$ denotes the probability density function (pdf) of $\mathcal{N}(0, I\sigma^2)$. We will often think of $\gamma$ as the privacy parameter (instead of $\alpha$).

When we take the log of pdf, it is easy to see that Eq. (1) is equivalent to testing if:

$$|\mathrm{logpdf}(Z) - \mathrm{logpdf}(\tilde{g}_s - g_i)| \leq \gamma \,, \tag{2}$$

which is easily testable for all batches' gradients $g_i$ for $i = 1, 2, \ldots, m$ since the log-pdf of isotropic Gaussian can be computed efficiently.

## 3.3 ALGORITHM

Algorithm 1 provides a description of the proposed method. We initialize $\theta_0$ randomly and iterate for up to $S$ learning steps. In each step, we randomly partition the training data $D$ into $m$ roughly equal batches $B_1, \ldots, B_m$. But unlike SGD, we only pick a single seed batch $B_s$ among them uniformly at

---

**Algorithm 1** Plausibly Deniable Stochastic Gradient Descent (PD-SGD)

---

**Input:** Training dataset $D$, number of batches $m$, number of training steps $S$, loss function $\mathcal{L}(\cdot)$, privacy test parameters $(\gamma, T)$.
    **Initialize:** $\theta_0$ randomly
    **for** $i = 1, 2, \ldots, S$ steps **do**
        **Randomly split** $D$ **into** $\{B_1, \ldots, B_m\}$     // Equal size batches
        **Pick seed batch** $B_s$ **uniformly at random**
        $g_s \leftarrow \nabla_\theta \mathcal{L}(\theta_{i-1}, B_s)$         // Compute gradient on seed batch
        $\tilde{g}_s \leftarrow g_s + Z$ where $Z \sim \mathcal{N}(0, \sigma^2 I)$     // Compute noisy gradient
        $\tau_{\text{count}} \leftarrow 0$
        $\theta_i \leftarrow \theta_{i-1}$
        **for** $j \in [1, m]$ **do**         // Privacy test and parameter updates
            $g_j \leftarrow \nabla_\theta \mathcal{L}(\theta_{i-1}, B_j)$     // Compute gradient on batch $B_j$
            $\tau_j \leftarrow \mathbb{1}_{|\text{logpdf}(\tilde{g}_s - g_s) - \text{logpdf}(\tilde{g}_s - g_j)| \leq \gamma}$     // Is gradient plausible?
            $\tau_{\text{count}} \leftarrow \tau_{\text{count}} + \tau_j$
            **if** $\tau_{\text{count}} \geq T$ **then**         // Enough plausible alternative batches?
                $\theta_i \leftarrow \theta_{i-1} - \eta\,\tilde{g}_s$     // Update model parameters with $\tilde{g}_s$
                **Break**
            **end if**
        **end for**
    **end for**

---

random. We then compute the gradient vector of the loss with respect to the model parameters under seed batch $B_s$, which results in $g_s$, and add isotropic Gaussian noise with scale $\sigma$ on it to obtain noisy gradient $\tilde{g}_s$.

Evaluating the privacy test involves the computation of the other batches' gradients. For this, we count the number of unique batches that satisfy Eq. (2). We compare this quantity to the threshold $T > 1$. If the quantity is greater than or equal, then we update the model parameters $\theta_i$ with the noisy gradient $\tilde{g}_s$ (and exit the inner loop early). Otherwise, the update is never applied (keep $\theta_i = \theta_{i-1}$) (i.e., we discard the update) and continue to the next step.

**Privacy-Utility Tradeoff.** Rejections of the privacy test drive the privacy (and utility) of the model. In particular, if the test never rejects any candidate gradient updates, then Algorithm 1 is equivalent to (vanilla) SGD. Informally, we expect utility to be maximized when the rejection rate is near 0, and we expect privacy to increase as rejection rates increase. Critically, the privacy test must reject precisely those gradients from batches that would leak private information (e.g., those that would increase the vulnerability to membership inference). We show theoretically why this is guaranteed to happen in the next section. We also demonstrate experimentally that this happens in practice in Section 5.

**Algorithmic Complexity.** Compared to SGD, Algorithm 1 only performs at most a single update of model parameters in each step. This update only occurs if the privacy test passes and it requires computing up to $m$ batches' gradients. Checking Eq. (2) is reasonably efficient in practice so the main computational bottleneck is the gradients' computations. However, observe that when the rejection rate is expected to be low, the algorithm will often not need to compute all $m$ batches' gradients to pass the test. In experiments (supplementary materials) we find that although PD-SGD is slower than SGD, it is often much faster than DP-SGD for a single training step, in large part because it does not require calculating per-example gradients.

## 4 THEORETICAL JUSTIFICATION: WHY DOES PD-SGD PROTECT PRIVACY?

So far, we described the PD-SGD algorithm and explained its privacy test. The premise is that if we only ever apply parameter updates based on batch gradients' that are plausibly deniable, then privacy is protected. Viewed through this lens, Algorithm 1 provides an intuitive guarantee.

In this section, we go beyond this intuition and show that the privacy test provably prevents those updates that would leak private information.

## 4.1 WHY DO (SOME) BATCHES PASS THE PRIVACY TEST?

The privacy test rejects gradient updates that are not plausibly deniable. In the following, we show that batches with gradients that have a large $l_2$ norm compared to other batches' gradients are rejected with overwhelming probability. The reason for this is the deep mathematical connection between the Gaussian distribution and the $l_2$-norm, which has been explored in other contexts (Figueiredo, 2001; Evans & Stark, 2002; MacKay, 2003). More precisely, the probability of passing the test decreases exponentially as a function of increasing $l_2$ distance to the closest other batch's gradient.

Consider a seed batch $B_s$, its associated gradient $g_s$, and another batch $B_i$ with gradient $g_i$. Recall that a noisy candidate gradient $\tilde{g}_s = g_s + Z$ is plausibly deniable with respect to batch $B_i$ iff Eq. (1) holds. In other words, we denote plausibility (of $\tilde{g}_s$ with respect to some $g_i$) as the probability that Eq. (1) holds:

$$q(s, i) = \Pr\left[\alpha^{-1} \leq \frac{p(\tilde{g}_s - g_s)}{p(\tilde{g}_s - g_i)} \leq \alpha\right] ,$$

where the probability $q(s, i)$ is taken over the randomness of $Z \sim \mathcal{N}(0, \sigma^2 I)$. This probability only depends on batches $B_s$ and $B_i$. The following result shows that it only depends on the $l_2$-distance between the two gradients, i.e., $||g_s - g_i||_2$.

**Lemma 1.** *For any seed batch with gradient $g_s$ and any mini-batch with gradient $g_i$, let $d = ||g_s - g_i||_2^2$. The probability that Eq. (1) holds depends only $d$ and we have:*

$$q(d) = q(s, i) = \Pr\left(Y \in \left[\frac{d - \tilde{\gamma}}{2\sigma\sqrt{d}}, \frac{d + \tilde{\gamma}}{2\sigma\sqrt{d}}\right]\right) , \tag{3}$$

*where $Y \sim \mathcal{N}(0, 1)$ and $\tilde{\gamma} = 2\sigma^2\gamma$.*

Lemma 1 shows that $q(d)$ is exactly the probability that a standard normal variable takes a value in $[\frac{d - \tilde{\gamma}}{2\sigma\sqrt{d}}, \frac{d + \tilde{\gamma}}{2\sigma\sqrt{d}}]$ where $\tilde{\gamma} = 2\sigma^2\gamma$. We provide a proof in Appendix B.

Intuitively, for $a \gg b > 0$ the probability $\Pr(a - b \leq Y \leq a + b)$ can be reasonably approximated as $2b\phi(a)$ where $\phi(\cdot)$ is the standard normal pdf, and thus the probability falls exponentially fast with $a$.

The following results derived from tail bounds on Lemma 1 show that plausibility falls off **exponentially** fast with the $l_2$-norm $d$ whenever $d$ is sufficiently large with respect to $\tilde{\gamma}$. This immediately implies that **any highly anomalous candidate gradient** (i.e., gradient with large $l_2$-norm to all other mini-batch gradients) **will be rejected with high probability.**

**Lemma 2.** *For any seed batch with gradient $g_s$ and any mini-batch with gradient $g_i$, and let $d$ be defined as in Lemma 1. If $d > 2\sigma^2\gamma$, we have that:*

$$q(d) < C_{d, \gamma, \sigma} \cdot \exp\left(-\left[\frac{d^2 + \tilde{\gamma}^2}{8d\sigma^2}\right]\right) . \tag{4}$$

*where $C_{d, \gamma, \sigma} = \frac{\sqrt{2d}\sigma}{\sqrt{\pi}} \cdot \left[\frac{\exp\left(\frac{\gamma}{2}\right)}{d - \tilde{\gamma}} - \frac{(d + \tilde{\gamma}) \cdot \exp\left(-\frac{\gamma}{2}\right)}{((d + \tilde{\gamma})^2 + 4\sigma^2 d)}\right].$*

We defer the proof of Lemma 2 to Appendix B. We also provide a simple upper-bound in Corollary 1 which is also in Appendix B.

## 4.2 PRIVACY GUARANTEES & PARAMETER TUNING

Recall from Section 3.1 that privacy leakage results from including examples that distort the gradient. Lemma 2 implies that privacy leakage is guaranteed to be mitigated in the following sense. Any example causing a large distortion to the batch gradient, if included, will result in a failure to pass the privacy test with a high probability.

To see this observe the following. Consider an example within a batch that has a highly distorting impact on this batch's gradient $g_s^\star$ compared to the batch's gradient without this example $g_s$, i.e., $||g_s^\star - g_s||_2^2$ is large. If $g_s^\star$ is also anomalous with respect to all other mini-batch gradients, i.e.,

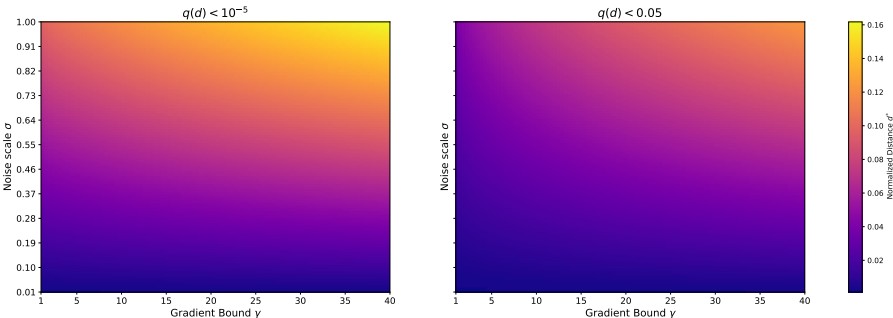

Figure 1: **Normalized Distance** $d^*$ **for varying** $\sigma$ **and** $\gamma$ **under different** $q(d)$. We observe that for a fixed probability of passing the test $q(d)$, the larger the product of $\sigma$ and $\gamma$ the larger the normalized distance $d^*$ can be, meaning that more anomalous batches pass the privacy test. Note that $d^* = \sqrt{d/k}$ where $k$ is the dimension of the gradient vector (we set $k = 7680$ for this case).

$d = \min_i ||g_s^\star - g_i||_2^2 \geq ||g_s^\star - g_s||_2^2$, then, the probability of passing the privacy test with threshold $T$ (assuming $T > 1$) is at most $(m - 1)q(d)$ by union bound.

Further, by tuning $\gamma$ and $\sigma$, we can make $q(d)$ arbitrarily small and therefore (in principle) eliminate the privacy leakage of any example. However, the relationships between $d$, $\sigma$ and $\gamma$ are complex. There is a tradeoff between $\sigma$ and $\gamma$ in terms of satisfying Eq. (1). Informally, for a fixed $\gamma$, the probability decreases exponentially with the ratio $\frac{d}{\sigma^2}$. So if $d$ is large then a large noise scale is required for plausibility (in which case privacy leakage is eliminated from the large noise). Conversely, with a small noise scale even relatively small deviations $d$ are not plausible.

To provide intuition and guide parameter tuning, we plot the minimum $d$ such that $q(d)$ is at most some $\delta > 0$ as a function of $\gamma$ and $\sigma$. This is shown in Fig. 1 for $\delta = 0.05$ and $\delta = 10^{-5}$, which plots $\sqrt{d/k}$, where $k$ is the dimension of the gradient vector (i.e., $g \in \mathbb{R}^k$) that used here for normalization. We observe that (as expected) we require larger $d^*$ for the same $\sigma$ and $\gamma$ for $q(d) < 10^{-5}$ compared to $q(d) < 0.05$. Moreover, for a fixed $q(d)$, the normalized distance $d^*$ appears to grow with the product of $\sigma$ and $\gamma$. This is consistent with Lemma 2, which suggests that the asymptotic behavior is driven by the product $\sigma^2 \gamma$. Furthermore, when tuning the privacy parameters, exploring combinations of $\sigma$ and $\gamma$ such that $\sigma^2 \gamma$ remains roughly constant is a sensible strategy.

## 5 EXPERIMENTS

### 5.1 SETUP

We use three of the most commonly used datasets for evaluating membership inference attacks (Shokri et al., 2017; Ye et al., 2022; Tang et al., 2022) and DP-SGD (De et al., 2022; Bao et al., 2024): CIFAR-10, CIFAR-100 and Purchase-100. For the models, we fine-tune ViT-B-16 for CIFAR-10 and CIFAR-100 following few-shot settings in (Tobaben et al., 2023) using 500 shots for CIFAR-10 and 1000 for CIFAR-100, linear model for Purchase-100, and Wide ResNet16-4 for CIFAR-10 training from scratch. We use the Privacy Meter toolbox [2] for the implementation of membership inference attacks. From it, we use the Population Attack (P-Attack), Reference Attack (R-Attack), Shadow model Attack (S-Attack) based on Ye et al. (2022) and Carlini et al. Attack (C-Attack) based on Carlini et al. (2022). We employ these four widely used attacks to comprehensively evaluate empirical privacy leakage and make fair comparisons between different methods. Note that our goal here is not to use the most exotic or recent attack, but to establish a fair empirical comparison between different defense methods, and thus we use a well-understood set of popular recent membership inference attacks. We provide more details in Appendix C.

Table 2: **Evaluations for PD-SGD**: We evaluate PD-SGD on three datasets with four different attacks. We report the average results and standard deviation among three independent runs. We can observe that PD-SGD can achieve a better privacy-utility trade-off than other empirical defense mechanisms and DP-SGD.

| Dataset | Method | Test acc | P-Attack | R-Attack | S-Attack | C-Attack |
|---|---|---|---|---|---|---|
| CIFAR-10 | Non-private | 96.09% (±0.02%) | 0.57(±0.01) | 0.69(±0.01) | 0.56 (±0.01) | 0.37% (±0.03%) |
| | AdvReg | 95.96% (±0.06%) | 0.56 (±0.01) | 0.59 (±0.01) | 0.55 (±0.00) | 0.31% (±0.01%) |
| | SELENA | 96.01% (±0.04%) | 0.55 (±0.00) | 0.51 (±0.01) | 0.56 (±0.02) | 0.33% (±0.02%) |
| | PD-SGD (param setting 1) | 96.18% (±0.06%) | 0.54 (±0.01) | 0.49 (±0.01) | 0.55 (±0.01) | 0.27% (±0.02%) |
| | PD-SGD (param setting 2) | 94.73% (±0.07%) | 0.53 (±0.01) | 0.49 (±0.01) | 0.53 (±0.01) | 0.20%(±0.03%) |
| | DP-SGD ($\varepsilon = 1$) | 68.97% (±0.11%) | 0.52(±0.01) | 0.50 (±0.01) | 0.52 (±0.01) | 0.17% (±0.01%) |
| | DP-SGD ($\varepsilon = 4$) | 93.53% (±0.07%) | 0.54 (±0.01) | 0.56 (±0.02) | 0.54 (±0.01) | 0.20% (±0.03%) |
| | DP-SGD ($\varepsilon = 8$) | 94.22% (±0.09%) | 0.54 (±0.00) | 0.59 (±0.01) | 0.54 (±0.01) | 0.23% (±0.02%) |
| CIFAR-100 | Non-private | 74.22% (±0.03%) | 0.73(±0.01) | 0.68(±0.01) | 0.73 (±0.01) | 0.38% (±0.03%) |
| | AdvReg | 72.08% (±0.03%) | 0.70(±0.01) | 0.68(±0.01) | 0.72 (±0.01) | 0.33% (±0.02%) |
| | SELENA | 68.46% (±0.04%) | 0.63(±0.00) | 0.60 (±0.01) | 0.65 (±0.01) | 0.19% (±0.02%) |
| | PD-SGD (param setting 1) | 72.56% (±0.06%) | 0.67(±0.01) | 0.62(±0.01) | 0.64 (±0.01) | 0.18% (±0.02%) |
| | PD-SGD (param setting 2) | 68.79% (±0.05%) | 0.62(±0.01) | 0.59 (±0.01) | 0.62 (±0.01) | 0.14% (±0.02%) |
| | DP-SGD ($\varepsilon = 1$) | 4.46% (±0.13%) | 0.50 (±0.01) | 0.50(±0.00) | 0.50 (±0.01) | 0.10% (±0.01%) |
| | DP-SGD ($\varepsilon = 4$) | 18.37% (±0.06%) | 0.50(±0.00) | 0.50 (±0.01) | 0.51 (±0.01) | 0.12% (±0.02%) |
| | DP-SGD ($\varepsilon = 8$) | 27.12% (±0.05%) | 0.51 (±0.01) | 0.52 (±0.01) | 0.51 (±0.01) | 0.13% (±0.03%) |
| Purchase-100 | Non-private | 68.56%(±0.12%) | 0.76(±0.01) | 0.78 (±0.01) | 0.77 (±0.01) | 0.12% (±0.02%) |
| | AdvReg | 57.56%(±0.07%) | 0.70(±0.01) | 0.70(±0.01) | 0.66 (±0.01) | 0.08%(±0.02%) |
| | SELENA | 64.31% (±0.09%) | 0.63(±0.00) | 0.73(±0.01) | 0.66 (±0.01) | 0.07%(±0.01%) |
| | PD-SGD (param setting 1) | 64.83% (±0.05%) | 0.63(±0.01) | 0.72(±0.01) | 0.64 (±0.01) | 0.06% (±0.01%) |
| | PD-SGD (param setting 2) | 61.16% (±0.07%) | 0.61(±0.01) | 0.59 (±0.02) | 0.60 (±0.01) | 0.06% (±0.01%) |
| | DP-SGD ($\varepsilon = 1$) | 22.51% (±0.22%) | 0.53(±0.01) | 0.54 (±0.01) | 0.54 (±0.00) | 0.04% (±0.01%) |
| | DP-SGD ($\varepsilon = 4$) | 43.46% (±0.15%) | 0.56 (±0.01) | 0.55(±0.01) | 0.56 (±0.01) | 0.07%(±0.02%) |
| | DP-SGD ($\varepsilon = 8$) | 47.61% (±0.12%) | 0.56(±0.00) | 0.56(±0.01) | 0.56 (±0.01) | 0.08% (±0.01%) |

Table 3: **Evaluate PD-SGD on ResNet-like model with Training from scratch:** Train WRN-16-4 from scratch with PD-SGD on CIFAR-10. We can observe the same thing: PD-SGD achieves a better privacy-utility trade-off than other defense mechanisms.

| Method | Test acc | P-Attack | R-Attack | S-Attack | C-Attack |
|---|---|---|---|---|---|
| Non-private | 87.22% (±0.13%) | 0.60 (±0.01) | 0.60 (±0.01) | 0.58 (±0.01) | 0.22% (±0.03%) |
| AdvReg | 75.38% (±0.09%) | 0.53 (±0.00) | 0.54 (±0.01) | 0.53 (±0.01) | 0.19% (±0.02%) |
| SELENA | 81.04% (±0.07%) | 0.53 (±0.01) | 0.53 (±0.01) | 0.53 (±0.01) | 0.19% (±0.01%) |
| PD-SGD (param setting 1) | 82.22% (±0.11%) | 0.53 (±0.01) | 0.52 (±0.01) | 0.51 (±0.01) | 0.19% (±0.01%) |
| PD-SGD (param setting 2) | 79.69% (±0.25%) | 0.53 (±0.00) | 0.50 (±0.01) | 0.51 (±0.01) | 0.15% (±0.01%) |
| DP-SGD ($\varepsilon = 1$) | 26.53% (±0.48%) | 0.50 (±0.00) | 0.49 (±0.01) | 0.50 (±0.01) | 0.07% (±0.02%) |
| DP-SGD ($\varepsilon = 4$) | 55.46% (±0.28%) | 0.50 (±0.01) | 0.49 (±0.01) | 0.50 (±0.01) | 0.10% (±0.01%) |
| DP-SGD ($\varepsilon = 8$) | 63.31% (±0.15%) | 0.51 (±0.01) | 0.50 (±0.00) | 0.51 (±0.01) | 0.13% (±0.02%) |

## 5.2 EVALUATIONS

We evaluate the utility and privacy of our proposed methods and other defense mechanisms. We primarily evaluate utility using the trained models' test accuracies, although we include results on computational overhead in Appendix D.1. We evaluate privacy using our selected set of four different membership inference attacks, namely P-Attack, R-Attack, and S-Attack, and C-Attack. For the first three, we report the attack AUC score. For C-Attack we report TPR at 0.01% FPR as advocated for by Carlini et al. (2022).

We use two sets of hyperparameters for PD-SGD. Parameter setting 1 is designed to optimize utility while maintaining reasonable privacy, while parameter setting 2 prioritizes better privacy at the cost of lower accuracy. Appendix E provides full details of the parameter settings.

Table 2 shows the results. We observe that PD-SGD, particularly with parameter setting 1, achieves **comparable utility** to non-private setting with a 96.15% test accuracy on CIFAR-10 and maintains robust performance on CIFAR-100 and Purchase-100, though slightly lower than some non-private baselines. Notably, PD-SGD exhibits stronger membership inference attack resilience than empirical defenses, with C-Attack performance being among the lowest recorded.

Furthermore, PD-SGD provides a favorable privacy-utility tradeoff even in cases where privacy is paramount (parameter setting 2). For instance, there is only approximately 7% decrease in test accuracy to obtain a reduction in attack AUC of nearly 0.16 for Purchase-100, compared to the non-private baseline.

Overall, findings show that PD-SGD achieves a superior trade-off between privacy and utility, surpassing empirical defenses. Compared to DP-SGD, the method sometimes provides good or better

---

[2]`https://github.com/privacytrustlab/ml_privacy_meter`

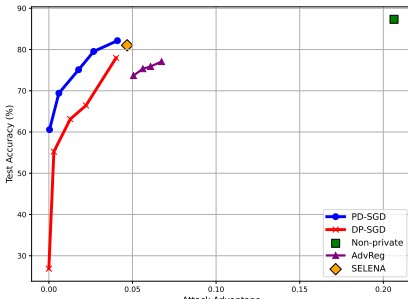

Figure 2: **Privacy-Utility Trade-off for different methods :** We train WRN-16-4 on CIFAR-10 from scratch with different defense methods. We can observe that PD-SGD provides a **better** Privacy-Utility trade-off than all other defense methods. Note that Attack Advantage is computed as $2 \times$ (Balanced Attack Accuracy $- 0.5$).

Table 4: **Impact of Privacy Test and Noise:** We keep all hyperparameters the same, only changing the threshold $T$ to control the privacy test. ✓means the presence of noise or the application of a privacy test, $\times$ means the absence of these components, and $\otimes$ represents the use of random rejection for gradient updates instead of standard privacy testing.

| Method | Noise | Privacy Test | Test acc | P-Attack | R-Attack | S-Attack | C-Attack |
|---|---|---|---|---|---|---|---|
| Non Private | $\times$ | $\times$ | 96.08% | 0.56 | 0.68 | 0.56 | 0.35% |
| Only Noise | ✓ | $\times$ | 94.99% | 0.54 | 0.57 | 0.55 | 0.30% |
| Only Privacy Test | $\times$ | ✓ | 96.01% | 0.55 | 0.56 | 0.56 | 0.32% |
| Random Rejection | ✓ | $\otimes$ | 94.78% | 0.55 | 0.54 | 0.54 | 0.28% |
| **PD-SGD** | ✓ | ✓ | **94.70%** | **0.53** | **0.48** | **0.53** | **0.20%** |

membership privacy but with higher test accuracy. For instance, PD-SGD provides both higher test accuracy and better MIA defense than DP-SGD for $\varepsilon = 8$ for CIFAR-10.

To demonstrate the generalizability of PD-SGD across different model architectures, we extend our evaluation to a ResNet-like architecture by training a Wide ResNet (WRN-16-4) model from scratch on the CIFAR-10 dataset. Table 3 shows the results. In this table, PD-SGD also exhibits a superior privacy-utility trade-off compared to alternative defense mechanisms. Notably, PD-SGD with parameter setting 1 achieves a test accuracy of 82.14%, surpassing other privacy-preserving methods such as SELENA (81.03%) and AdvReg (75.34%). Moreover, PD-SGD achieves a significantly lower vulnerability to membership inference attacks. In particular, the R-Attack AUC score shows a marked decrease from 0.60 to 0.51 with parameter setting 2 of PD-SGD.

We illustrate the privacy-utility tradeoff between methods visually in Fig. 2. The x-axis shows the attack advantage and the y-axis shows the test accuracy for the WRN-16-4 model trained on CIFAR-10. Compared to DP-SGD, PD-SGD provides higher test accuracy for the same attack advantage. Compared to empirical defenses, a major advantage of PD-SGD is that it offers a way to navigate the tradeoff (through the privacy parameter) and not (only) a fixed point on the privacy-utility landscape.

## 6 ABLATION STUDY: WHY DOES PD-SGD WORK?

In this section, we perform a set of ablation experiments to examine the effect of each component within PD-SGD. We also explore why PD-SGD effectively protects privacy.

In Appendix D, we explore trade-offs between the privacy parameters, discuss parameter tuning, and provide additional experiments such as computation time per training step.

### 6.1 HOW PRIVACY TEST AND NOISE HELPS DEFEND MIA?

Compared to (vanilla-)SGD, PD-SGD includes two components: (1) noise addition to the seed batch's gradient, and (2) a plausible deniability-based privacy test. We create a set of principled experiments to isolate the effect of these two components.

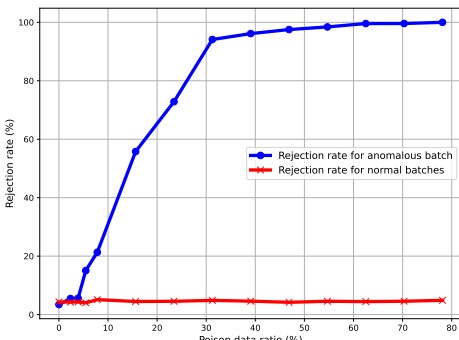

Figure 3: **Rejection rate for anomalous and normal batches.** Rejection rate for the anomalous batch increases to close to 100% as the proportion of poisoned examples increases while the rate for normal batches remains stable. This suggests that, as desired for privacy and utility, only those gradient updates that may cause privacy leaks are rejected.

- **Only Noise:** we set the threshold $T = 1$, guaranteeing the privacy test will always pass.
- **Only Privacy Test:** use privacy test normally, but update parameters using the un-noised gradient.
- **Random Rejection:** seed batches' gradients are randomly rejected at the same rate as PD-SGD.

Table 4 shows the results. Adding noise to the gradient without the privacy test does not effectively defend against membership inference. The R-Attack success rate decreases substantially, but there is no substantial decrease for P-Attack, S-Attack, and C-Attack. Similarly, if the privacy test is used but the gradient is un-noised or if updates are randomly rejected, we again see no major decrease in membership inference attack success rates. By contrast, PD-SGD exhibits the largest effect in mitigating membership inference attacks. The R-Attack success rate drops further to 0.48, and other attack vectors like P-Attack, S-Attack, and C-Attack are similarly reduced.

These results demonstrate that it is the combination of both noise addition and privacy test that results in the observed privacy protection of PD-SGD.

### 6.2 REJECTION OF ANOMALOUS BATCHES

How do we know that PD-SGD rejects gradient updates from anomalous batches and only those from anomalous batches? We intentionally generate anomalous batches to evaluate this by flipping the labels of a subset of examples ("poisoned examples") and grouping them into a single batch with other normal samples. We ensure that throughout training the poisoned examples are in the same "anomalous" batch. We then collect the rejection rates when the anomalous batch is the seed and when other batches are the seed, for varying proportion of poisoned examples.

Results are shown in Fig. 3, where we observe that for normal batches remain consistently low, as expected and desired. This means that the privacy test does not discard updates unnecessarily. However, when the anomalous batch is selected as seed, the rejection rate increases significantly and quickly plateaus near 100% as the proportion of poisoned examples increases. This indicates that PD-SGD effectively identifies and rejects anomalous batches, preventing the model parameters from being updated in such cases.

## 7 CONCLUSIONS

We proposed PD-SGD, a new approach for private learning without compromising performance. PD-SGD is based on a rejection sampling approach using a privacy test. Theoretical and experimental results demonstrate that PD-SGD provides a superior privacy-utility trade-off compared to both existing methods with provable privacy such as DP-SGD and empirical defenses. This makes PD-SGD a promising solution for enhancing privacy protection in practical deep-learning applications.

ETHICS STATEMENT

This paper proposes a new approach to protect privacy when training deep learning models. Protecting privacy when deploying machine learning is important because it has the potential to substantially mitigate harms to the privacy of individuals. However, it is worth noting that (some) existing research work suggests that the privacy benefits of some technical approaches may not be shared equally among all groups and individuals, and could therefore potentially lead to unfairness.

REPRODUCIBILITY STATEMENT

To help with reproducibility, we provide full details of the experiments setup in Appendix C including the hyperparameters we used in Appendix E. Due to organizational restrictions, we are unable to release the source code publicly or open source it. Therefore, given the open nature of the ICLR review process, we could not include the code as part of the supplementary materials. However, upon request, we can provide a private link to an anonymous repository for reviewers and ACs only.

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

# A  SYMBOLS

Table 5: Table of Symbols.

| Symbol | Meaning | Where |
|--------|---------|-------|
| $(x, y)$ | Individual Example From Training Set | Section 2.1 |
| $\theta$ | Model Parameter Vector — $\theta \in \mathbb{R}^k$ | Section 2.1 |
| $B_i$ | SGD Mini-Batch $i$ | Section 3.2 |
| $g_s$ | Gradient (of the Loss wrt $\theta$) of Batch $i$ | Section 3.2 |
| $B_s$ | Chosen "Seed" Batch | Section 3.2 |
| $g_s$ | Gradient of Seed Batch | Section 3.2 |
| $\tilde{g}_s$ | Noisy Gradient (of Seed Batch) | Section 3.2 |
| $\sigma$ | Privacy Parameter — Noise Scale | Section 3.2 |
| $Z$ | Gaussian Noise — $\mathcal{N}(0, \sigma^2 I)$ | Section 3.2 |
| $\gamma$ | Privacy Parameter — Log-PDF Threshold | Section 3.3 |
| $T$ | Privacy Parameter — Plausible Batches Threshold | Section 3.3 |
| $\tilde{\gamma}$ | (Half-)Width of Acceptance Region — $\tilde{\gamma} = 2\sigma^2\gamma$ | Section 4.1 |
| $d$ | Squared $l_2$ Distance of Gradients Between Seed and Batch $i$ — $d = \|g_s - g_i\|_2^2$ | Section 4.1 |
| $q(d)$ | Probability that Eq. (1) holds for a given $d$ | Section 4.1 |

# B  PROOFS

We now prove Lemma 1.

*Proof of Lemma 1.* Consider the ratio of probabilities bounded by Eq. (1) and expand using the Gaussian PDF. We get:

$$\frac{p(\tilde{g}_s - g_s)}{p(\tilde{g}_s - g)} = \frac{\exp\left(-(2\sigma^2)^{-1} \sum_{j=1}^{k} Z_j^2\right)}{\exp\left(-(2\sigma^2)^{-1} \sum_{j=1}^{k} (Z_j + (g_{s,j} - g_{i,j}))^2\right)}$$

$$= \exp\left(-(2\sigma^2)^{-1} \sum_{j=1}^{k} \left[Z_j^2 - (d_j + Z_j)^2\right]\right)$$

$$= \exp\left(-(2\sigma^2)^{-1} \left[-d - 2\sum_{j=1}^{k} d_j Z_j\right]\right),$$

where $d_j = g_{s,j} - g_{i,j}$ and $d = \sum_{j=1}^{k} d_j^2 = \|g_s - g_i\|_2^2$.

Plugging this into the inequality, taking the log and some reorganization we get that the candidate gradient is plausibly deniable with respect to $g_i$ iff:

$$-\frac{\gamma\sigma}{\sqrt{d}} \leq \frac{\sqrt{d}}{2\sigma} + \sum_{j=1}^{k} \frac{d_j}{\sqrt{d}} \frac{Z_j}{\sigma} \leq \frac{\gamma\sigma}{\sqrt{d}} .$$

Since $Z_j \sim \mathcal{N}(0, \sigma^2)$, the summand for $j$ is distributed as $\mathcal{N}(0, d^{-1}d_j^2)$. Further, since the sum of i.i.d. Gaussian random variable is distributed a Gaussian random variable with the sum of the means and the sum of the variance, we recognize that $Y = \sum_{j=1}^{k} \frac{d_j}{\sqrt{d}} \frac{Z_j}{\sigma} \sim \mathcal{N}(0, 1)$.

Thus reducing the plausibility of a candidate gradient to:

$$\frac{\sqrt{d}}{2\sigma} - \frac{\gamma\sigma}{\sqrt{d}} \leq Y \leq \frac{\gamma\sigma}{\sqrt{d}} + \frac{\sqrt{d}}{2\sigma} , \tag{5}$$

and further to

$$\frac{d - 2\gamma\sigma^2}{2\sigma\sqrt{d}} \leq Y \leq \frac{d + 2\gamma\sigma^2}{2\sigma\sqrt{d}} \tag{6}$$

where we have used symmetry so that $-Y$ has the same distribution as $Y$.

Therefore, $Y$ needs to be within a band of width $\frac{\tilde{\gamma}}{\sigma\sqrt{d}}$ around $\sqrt{d}/2\sigma$ where $\tilde{\gamma} = 2\sigma^2\gamma$, which completes the proof. □

The proof of Lemma 2 relies on the following standard normal upper and lower tail bounds:

**Lemma 3.** *Let $X \sim N(0,1)$. For $t > 0$, we have:*

$$\frac{t}{t^2+1}(\sqrt{2\pi})^{-1}\exp\left(-t^2/2\right) < \Pr(X > t) < (t\sqrt{2\pi})^{-1}\exp\left(-t^2/2\right).$$

Note that tighter bounds are available (Cook (2024); Duembgen (2010)).

*Proof of Lemma 2.* Let $a = \frac{\sqrt{d}}{2\sigma}$ and $b = \frac{\gamma\sigma}{\sqrt{d}}$. We have from Lemma 1 that $q(s,i) = \Pr(a - b \leq X \leq a + b)$ for $X \sim N(0,1)$. Thus:

$$q(s,i) = \Pr(X > a - b) - \Pr(X > a + b)$$

$$< \frac{1}{(a-b)\sqrt{2\pi}}e^{-(a-b)^2/2} - \frac{(a+b)}{((a+b)^2+1)\sqrt{2\pi}}e^{-(a+b)^2/2}$$

$$= \frac{1}{\sqrt{2\pi}}\left[\frac{1}{a-b}e^{-(a-b)^2/2} - \frac{(a+b)}{(a+b)^2+1}e^{-(a+b)^2/2}\right]$$

$$= \frac{e^{\frac{-(a^2+b^2)}{2}}}{\sqrt{2\pi}}\left[\frac{e^{ab}}{a-b} - \frac{(a+b)}{(a+b)^2+1}e^{-ab}\right].$$

Substituting back $a$ and $b$ in terms of $d, \sigma, \gamma$ yields the result. □

The following corollary of the lemma provides a simple upper bound whenever $d > \tilde{\gamma}$.

**Corollary 1.** *Let $d \geq \frac{\tilde{\gamma}}{f}$ for some $0 < f < 1$. Then:*

$$q(d) < \frac{e^{-(\frac{d}{8\sigma^2}+\frac{\gamma^2\sigma^2}{2d})}}{\sqrt{2\pi d}}2\sigma\left[\frac{e^{\gamma/2}}{1-f} - \frac{e^{-\gamma/2}}{2+f}\right] \quad (7)$$

*Proof of Corollary 1.* Let $d \geq 2\gamma\sigma^2$ which implies $a - b \geq 0$. When $d$ increases, $a$ increases but $b$ decreases. So, we can bound $a - b$ and $a + b$ as follows:

Suppose $b \leq fa$ where $0 \leq f < 1$ and $a > 1$, then

$$\frac{1}{a-b} \leq \frac{1}{a(1-f)}$$

$$\frac{a+b}{(a+b)^2+1} \geq \frac{1}{a(2+f)}$$

Based on this, we can get:

$$q(s,i) < \frac{e^{\frac{-(a^2+b^2)}{2}}}{\sqrt{2\pi}}\left[\frac{e^{ab}}{a-b} - \frac{(a+b)}{(a+b)^2+1}e^{-ab}\right]$$

$$< \frac{e^{\frac{-(a^2+b^2)}{2}}}{\sqrt{2\pi}a}\left[\frac{e^{ab}}{1-f} - \frac{e^{-ab}}{2+f}\right].$$

Observe that $ab = \gamma/2$, $a^2 = \frac{d}{4\sigma^2}$, $b^2 = \frac{\gamma^2\sigma^2}{d}$

So:

$$q(s,i) < \frac{e^{-(\frac{d}{8\sigma^2}+\frac{\gamma^2\sigma^2}{2d})}}{\sqrt{2\pi d}}2\sigma\left[\frac{e^{\gamma/2}}{1-f} - \frac{e^{-\gamma/2}}{2+f}\right].$$

□

## C    EXPERIMENTS SETUP

### C.1    DATASETS

We use the three of the most commonly used datasets for evaluating membership inference attacks (Shokri et al., 2017; Ye et al., 2022; Tang et al., 2022) and DP-SGD (De et al., 2022; Bao et al., 2024).

**CIFAR-10**  (Krizhevsky et al., 2009) contains 60,000 images with 10 classes. We use 50,000 as the full training set and 10,000 as the test set as most papers do. Each example has three RGB channels and size $32 \times 32$ pixels. For fine-tuning tasks, we only use 500 data samples for training and 30,000 for training from scratch.

**CIFAR-100** is a well-known benchmark in the field of computer vision, also collected by Krizhevsky et al. (2009). CIFAR-100 contains 60,000 color images, each with a resolution of $32 \times 32$ pixels. It is more complex than the CIFAR-10 dataset; the images are organized into 100 distinct classes. The dataset allocation includes 50,000 images for training purposes and 10,000 for testing. For finetuning task, we only use 1000 data samples for training and the rest of training data examples are used for MIA evaluation. For training from scratch, we use 25,000 data samples as the same setting in Zarifzadeh et al. (2024).

**Purchase-100** is based on Kaggle's "acquire valued shoppers" challenge[3] and processed and simplified as introduced in Shokri et al. (2017). The dataset contains shopping records for thousands of individuals and includes 197,324 data entries. For training, we use 25,000 samples and the rest for testing. For MIAs, we use 25,000 samples from test set as shadow dataset.

### C.2    MODELS

**Vit-B-16** are pre-trained on the LAION-2B dataset (Schuhmann et al., 2022). We obtain the model from Open Clip[4] and add a linear layer as a classification head. We only fine-tune this last layer and freeze the weights of other layers. We utilize this model for CIFAR-10 and CIFAR-100 fine-tuning tasks.

**Wide ResNet (WRN)** (Zagoruyko & Komodakis, 2016) is a popular variant of the ResNet (Residual Network) model (He et al., 2016). The architecture increases the number of channels in convolutional layers (width) rather than the number of layers (depth). We use WRN-16-4 in experiments which is also commonly used in many DP-SGD related work (Bao et al., 2024; De et al., 2022; Sander et al., 2023). We train the model from scratch on CIFAR-10. We use WRN-28-2 for training from scratch on CIFAR-100.

**Linear model** is commonly used for tabular data such as Purchase-100. We use this one-layer linear model for experiments on Purchase-100.

### C.3    SETUPS

We implemented PD-SGD using PyTorch. For DP-SGD, we use Opacus (Yousefpour et al., 2021). For other empirical defense mechanisms, we reproduce them using SELENA's (Tang et al., 2022) original code-base[5]. For membership inference attack, we use the Privacy Meter toolbox [6]. From it, we use Population Attack (P-Attack), Reference Attack (R-Attack), Shadow model Attack (S-Attack) based on Ye et al. (2022) and Carlini et al. Attack (C-Attack) based on Carlini et al. (2022). We employ these four widely used attacks to comprehensively evaluate empirical privacy leakage and make fair comparisons between different methods. Note that our goal here is not to use the most exotic or recent attack, but to establish a fair empirical comparison between different defense methods, and thus we use a well-understood set of popular recent membership inference attacks.

---

[3]https://kaggle.com/c/acquire-valued-shoppers-challenge/data
[4]https://github.com/mlfoundations/open_clip
[5]https://github.com/inspire-group/MIAdefenseSELENA
[6]https://github.com/privacytrustlab/ml_privacy_meter/tree/173d4ad80f183ae6e1867b2793dfffe0633107d0

Table 6: **Computational Time per step:** We measure the GPU time for SGD, DP-SGD, and our proposed PD-SGD for one step with the same model and the same amount of data. We report the average time among 3 steps. For CIFAR-10 (Finetuning), we use Vit model and for CIFAR-10 (From scratch), we train WRN-16-4 from scratch. We can observe that although PD-SGD is slower than SGD, it takes less time than DP-SGD.

| Dataset | Method | Time (ms) |
|---|---|---|
| CIFAR-10 (Finetuning) | DP-SGD | 18.86 ($\pm$0.08) |
| | PD-SGD | 7.70 ($\pm$0.10) |
| | SGD | 0.49 ($\pm$0.03) |
| CIFAR-10 (From scratch) | DP-SGD | 2492.11 ($\pm$8.06) |
| | PD-SGD | 1780.16 ($\pm$15.72) |
| | SGD | 344.47 ($\pm$0.20) |

**Details for Attacks:** We keep the same attack setting for all defense mechanisms for a fair comparison. For all datasets, other than the part we used for training the target models, the rest of training samples are used as shadow datasets for shadow models or reference models. For all shadow models or reference models, we sample the same amount of data samples as target dataset for training. We use 8 shadow models for S-Attack, R-Attack and C-Attack. For the C-Attack, we use the online version of it and adopted from privacy meter.[7] When evaluating attack, we always use balanced evaluation dataset (50% member and 50% non-member). When reporting (balanced) accuracy, we always select the threshold with the highest attack accuracy.

**Details for Defenses:** We keep the same parameter setting for all other empirical defense mechanisms as SELENA's original code-base. For DP-SGD, we set the clipping threshold to 1 and use the same batch size as PD-SGD and SGD. We also perform a hyperparameter search to identify the best learning rate for every run.

## D  ADDITIONAL EXPERIMENTS

### D.1  COMPUTATIONAL TIME MEASUREMENT

We evaluate the running time of PD-SGD for one training step. We conduct experiments using CIFAR-10 by fine-tuning the ViT model, following the same setup as described for Table 2. We also train the WRN-16-4 model from scratch following the same setting in Table 3. The time is averaged over three consecutive steps taken from the middle of the training process. For comparison, we also measure the time of standard SGD and DP-SGD under the same conditions. The results are summarized in Table 6. As demonstrated, PD-SGD is noticeably slower than standard SGD but notably faster than DP-SGD for a single training step. However, the total training time also depends on the algorithm's convergence rate, which we leave the analysis of for future work.

### D.2  UNDERSTANDING PARAMETERS OF PD-SGD

Table 7: **Impact of $\gamma$**

| $\gamma$ | Test Acc | Reject Rate | Best Attack |
|---|---|---|---|
| 1 | 92.78% | 99.54% | 0.52 |
| 2 | 94.70% | 30.31% | 0.53 |
| 3 | 94.71% | 13.70% | 0.56 |
| 4 | 94.74% | 5.78% | 0.57 |
| 6 | 94.80% | 2.25% | 0.59 |

Table 8: **Impact of $\sigma$**

| $\sigma$ | Test Acc | Reject Rate | Best Attack |
|---|---|---|---|
| 0.1 | 17.19% | 99.95% | 0.52 |
| 0.15 | 96.02% | 0.15% | 0.54 |
| 0.2 | 95.70% | 0.03% | 0.55 |
| 0.4 | 93.67% | 0.00% | 0.55 |
| 1.0 | 85.23% | 0.00% | 0.56 |

Table 9: **Impact of $T$**

| $T$ | Test Acc | Reject Rate | Best Attack |
|---|---|---|---|
| 1 | 64.78% | 0.00% | 0.76 |
| 2 | 64.81% | 10.17% | 0.75 |
| 3 | 64.76% | 18.86% | 0.71 |
| 5 | 62.66% | 84.68% | 0.64 |
| 7 | 3.21% | 99.90% | 0.50 |

Recall that PD-SGD has three parameters — $\sigma$, $\gamma$, and $T$ — that control the privacy-utility trade-off. In this section, we discuss how these parameters impact the performance of PD-SGD.

---

[7]https://github.com/privacytrustlab/ml_privacy_meter/tree/173d4ad80f183ae6e1867b2793dfffe0633107d0/benchmark

We first fine-tune the ViT model on CIFAR-10 with different $\gamma$ values while keeping all other parameters fixed. The results are presented in Table 7. We observe that as $\gamma$ decreases, the model's test accuracy experiences a slight decline. However, the Best Attack AUC diminishes substantially. Notably, when $\gamma$ decreases from 2 to 1, even though the Best Attack AUC decreases slightly, the reject rate increases sharply to 99.54%, and the test accuracy drops to 92.78%. This suggests that $\gamma = 2$ may be the optimal choice for this parameter setting.

We perform similar experiments with different $\sigma$ values and present the results in Table 8. We observe that when $\sigma$ is large (i.e., $\sigma > 0.2$), the gradients can easily pass the Privacy Test, but the Best Attack AUC remains high, and the model fails to achieve good test accuracy due to the large noise introduced during training. When $\sigma$ is relatively small, although some gradients are rejected, it provides better defense performance (lower Attack AUC). However, if $\sigma$ is too small, such as 0.1, under the same $\gamma$ and $T$, it becomes very difficult for gradients to pass the privacy test, resulting in low test accuracy.

We also test different $T$ values while keeping all other parameters fixed. We train the linear model on Purchase-100 and present the results in Table 9. We observe that as $T$ increases, it becomes harder for gradients to pass the privacy test. Consequently, the reject rate increases, test accuracy decreases, but better defense performance is achieved (lower Attack AUC).

Therefore, based on these tables and results, we find that the observations corroborate our findings in Fig. 1. This demonstrates that PD-SGD can provide a wide range of privacy-utility trade-offs through different parameter settings. On the other hand, to achieve a better privacy-utility trade-off, it is advisable to tune all three parameters together rather than adjusting only one parameter.

Table 10: **Impact of batch size on Purchase-100 and CIFAR-10**

| Dataset | Batch size | Test Acc | Reject Rate | Best Attack |
|---|---|---|---|---|
| Purchase-100 | 1024 | 0% | 100% | 0.5 |
| | 2048 | 60.10% | 88.94% | 0.62 |
| | 3072 | 64.76% | 10.41% | 0.73 |
| | 4096 | 64.80% | 9.06% | 0.74 |
| | 5120 | 64.73% | 0% | 0.77 |
| CIFAR-10 | 1024 | 60.24% | 55.85% | 0.51 |
| | 2048 | 74.35% | 37.63% | 0.51 |
| | 3072 | 80.40% | 20.07% | 0.53 |
| | 4096 | 80.59% | 14.07% | 0.53 |
| | 5120 | 81.57% | 7.41% | 0.54 |

### D.3 UNDERSTANDING BATCH SIZE IN PD-SGD

The batch size plays an important role in terms of privacy. There are extreme edge cases that are unrealistic where the batch size is the entire training set or the batch size is a single example. For more realistic batch sizes there are several tradeoffs and ultimately the behavior depends also on the chosen privacy parameters.

We conduct experiments on Purchase-100 and CIFAR-10 to further understand the batch size in PD-SGD and report results in Table 10. Results indicate that as batch size increases, the rate of deniability typically decreases—larger batches more easily pass the privacy test (for fixed privacy parameters) due to the averaging effect you described across different datasets. However, this does not necessarily translate into better privacy protection, as the potential for individual sample contributions to still be inferred remains.

Moreover, we found that adjusting other parameters—e.g., $\sigma$, $\gamma$, and threshold can help mitigate these effects, maintaining a balance between utility and privacy across varying batch sizes. For example, for the batch size = 1024, if we double the $\gamma$, we can decrease the reject rate to 56.98% and achieve a test accuracy of 63.87% with Best Attack AUC of 0.68.

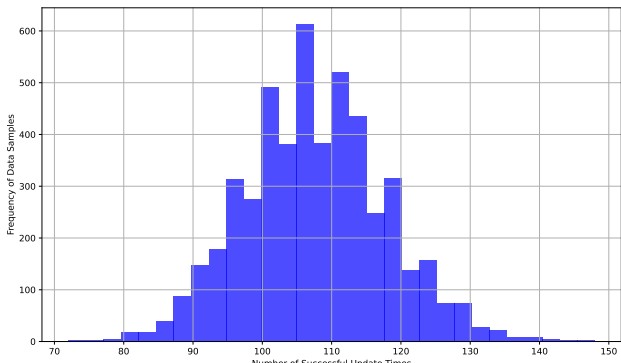

Figure 4: **Distribution of data samples' successful update** Histogram of all training data samples' successful update. The average count is 107.53 ($\pm$10.27) and the min and max are 72 and 148, respectively.

These results underscore the importance of carefully tuning all parameters in relation to batch size to uphold robust privacy guarantees while preserving utility.

Table 11: **Evaluate PD-SGD on CIFAR-100 for training from scratch**

| Method | Test Acc | P-Attack | R-Attack | S-Attack | C-Attack |
|---|---|---|---|---|---|
| Non-Private | 56.27% | 81.71% | 81.91% | 81.85% | 0.37% |
| PD-SGD(param 1) | 53.63% | 58.80% | 52.81% | 57.46% | 0.15% |
| PD-SGD(param 2) | 47.07% | 54.27% | 50.56% | 50.00% | 0.12% |
| DP-SGD($\varepsilon = 8$) | 18.24% | 52.29% | 49.58% | 51.03% | 0.11% |

## D.4 TRAIN FROM SCRATCH ON CIFAR-100

We used small training set sizes for these experiments to ensure the resulting models would be vulnerable to MIA so that it would be clear if the desired level of protection was indeed achieved. However, we also included other experiments in our paper where we used much larger training set sizes (e.g., Table 3). In addition, we conducted further experiments using a larger subset of CIFAR-100. We follow the experiment setting in Zarifzadeh et al. (2024) which trains a WRN-28-2 from scratch on 25k samples of CIFAR-100. We show the results in Table 11. It can be observed that PD-SGD can successfully defend different MIA attacks for example Attack AUC is decreased significantly from around 81% to 54% by using param setting 2 of PD-SGD. Compared to DP-SGD, PD-SGD provides much better utility.

## D.5 FREQUENCY OF EXAMPLES USED OF PD-SGD

Since PD-SGD works by rejecting implausible gradient updates, some training set examples may be used more frequently to update parameters than others. To investigate this, we record the successful update counts for each data sample in the training set in a case where parameters are set to achieve roughly 15% reject rate. We show this distribution in Fig. 4. We can observe that as expected there is a range of update frequencies. However, no training set example is used fewer than 72 times, so no example is systematically excluded from influencing the final model.

Table 12: Impact of Clip Threshold of DP-SGD

| Clip Threshold | Test Acc | P-Attack | R-Attack | S-Attack | C-Attack |
|:---:|:---:|:---:|:---:|:---:|:---:|
| 0.1 | 93.49% | 0.54 | 0.56 | 0.54 | 0.18% |
| 1 | 93.56% | 0.54 | 0.56 | 0.54 | 0.18% |
| 10 | 93.54% | 0.54 | 0.57 | 0.54 | 0.20% |

Table 13: Hyperparamters setting for experiments in Table 2 and Table 3

| Dataset | Param setting | $\sigma$ | $\gamma$ | T | Step | Reject Rate |
|:---:|:---:|:---:|:---:|:---:|:---:|:---:|
| CIFAR-10 | 1 | 0.1 | 40 | 2 | 20000 | 27.78% |
|  | 2 | 0.3 | 2 | 3 | 20000 | 30.31% |
| CIFAR-100 | 1 | 0.1 | 50 | 3 | 20000 | 44.08% |
|  | 2 | 0.2 | 10 | 3 | 20000 | 46.35% |
| Purchase-100 | 1 | 0.01 | 1000 | 3 | 100000 | 3.91% |
|  | 2 | 0.01 | 750 | 3 | 100000 | 87.76% |
| CIFAR-10 (for Table 3) | 1 | 0.01 | 40000 | 3 | 100000 | 1.06% |
|  | 2 | 0.02 | 7000 | 3 | 100000 | 3.70% |
| CIFAR-100 (for Table 11) | 1 | 0.01 | 100000 | 3 | 10000 | 0.49% |
|  | 2 | 0.01 | 9000 | 3 | 10000 | 32.81% |

## D.6 IMPACT OF CLIP THRESHOLD OF DP-SGD

To further investigate the impact of the clipping threshold in DP-SGD on privacy protection, we fixed all other parameters and varied the clipping threshold, as shown in Table 12. We can observe that even though the clip threshold changes, the model's utility and privacy are almost the same. However, during these experiments, we do find that if the clip threshold is changed, the learning rate also needs to be tuned properly to get the optimal utility. It makes sense that the impact on privacy of the clipping threshold should not be substantial since in DP-SGD the noise added to the gradient is scaled by the clipping norm.

## E PRIVACY HYPERPARAMETERS TUNING

Table 13 shows the hyperparameters settings we used for Table 2 and Table 3.

There are two broad strategies for tuning the privacy (hyper)parameters: (1) leverage the theoretical insights from Section 3.1; or (2) rely on empirically successful heuristics.

**Theory-based strategy:** As explained in Section 3.1, by tuning $\sigma$ and $\gamma$, we can make $q(d)$ arbitrarily small. If we have a desired bound on $d$, then we can find combinations of $\sigma$ and $\gamma$ that achieve the desired effects (e.g., see Fig. 1). This can for example be done through a grid search.

**Empirical strategy:** Alternatively, we found that the following two-steps strategy is easy to follow and yields good trade-offs. Step 1: tune the noise $\sigma$ to achieve acceptable utility, ignoring the privacy test. This helps determine an upper limit for utility. Step 2: tune $\gamma$ and the threshold $T$, which allows for fine-grained control over the privacy-utility trade-off. We used this two-step sequential tuning approach in our experiments.

A useful heuristic while tuning $\gamma$ and $T$ is to monitor the rejection rate. However, note that there exists favorable trade-offs for a wide-range of rejection rates, and a useful rule of thumb is therefore only to avoid extreme values (e.g., 0% — no privacy guarantee; 100% — no utility / full privacy).

## F LIMITATIONS & FUTURE RESEARCH

PD-SGD provides favorable privacy-utility tradeoffs compared to alternative methods both empirical and DP-SGD. It also provides a guarantee that anomalous batches are provably rejected with high probability. However, it is not meant as a direct replacement for DP-SGD, since the guarantees are

different. Working within the differential privacy framework is advantageous due to properties such as composition and post-processing.

Note that we intend our technique to mostly be used in a centralized learning environment where adversaries only observe the final model weights (or run inference with the trained model as a black box). We assume that the full training transcript (i.e., gradient updates, intermediate updates, whether the privacy test passes) is not accessible to the adversary. This assumption may prevent PD-SGD from being used in some settings such as federated learning.

PD-SGD has the advantage of providing better utility. It also starts a promising direction of future research in learning mechanisms that use privacy tests to enforce desirable properties. Further exploration of the theoretical properties such as composition, bounds on membership inference success rates, fairness considerations; and practical implications of PD-SGD is left for future work.

