# OpenReview forum: "Deep Learning with Plausible Deniability"
_ICLR.cc/2025/Conference — Submitted to ICLR 2025_

### Official Review · Reviewer_cMXB · 2024-10-25

**Soundness:** 3
**Presentation:** 4
**Contribution:** 3
**Rating:** 6
**Confidence:** 4

**Summary:**

This work proposes a novel method for training private ML models using the plausibly deniable SGD. The work shows that by discarding the gradients that are not plausibly deniable, the users can obtain quantifiable guarantees of privacy and better empirical performance than the standard DP methodology (DP-SGD).

**Strengths:**

The approach is novel and really interesting. The work itself is well-motivated, easy-to-follow and well-written. Most of the details are sound to me and the empirical results are mostly promising. There is a large number of evaluation with respect to attacks, which is always great to see.

**Weaknesses:**

To start with, I am not entirely sure on how sound the algorithm is. Firstly, it is not clear what exactly happens if the gradient is implausible - do you remove it or convert all values to 0's or something else entirely? Because 'removal' in a normal sense of a word would result in different sampling for the rest of the users and would cost them more epsilon, making some of them more vulnerable (which has knock-on effects on disparate vulnerability to MIA and how the method affects that). Secondly, this has an effect on the fairness of the algorithm: the benchmarks you used only consider the standard (somewhat easy and mostly balanced) datasets. In order to see how the method performs with respect to privacy attacks and subgroup utility, more analysis needs to be done and I request that the authors add a section on how performance imbalance is affected under PD-SGD.

There is also an issue with the way the paper is positioned. I admit I had to re-read the work because it is not immediately clear this method is, in fact, not designed to be differentially private (primarily because of the discussion on adding the gaussian noise and the privacy guarantees afterwards). Just to make it more clear to the reader, I would suggest you point this out early in the text. Especially since following claim only gets introduced in 2.4: 'With some additional randomisation this can be made epsilon-delta DP' What is this additional randomisation and where does it come from? Where is it discussed and how do you convert the guarantees? This sounds (in crude terms) a little bit like k-anonymity here, where you have a rejection sampling based on the number of rows that could have been used to produce the synthetic row (i.e. a probabilistic notion of k). Could you elaborate on why this is/isnt the case and what exactly needs to be done to establish the mapping between the methods?

I am not fully convinced on your choice of defence mechanisms. You seem to have picked Selena, which makes sense, but follow-up works have been proposed which significantly improved the effectiveness of the method [1] and since this method is empirical, I wonder why you did not evaluate PATE, which actually does come with privacy guarantees and is based on a similar method? If you are only doing training-based methods, then DKFD could be a good one to consider? AdvReg also does not seem like a strong candidate to compare against since it was already shown (in the original paper) to be mostly effective for data with a small number of classes (i.e. not cifar-100), making some of the comparisons less meaningful. Finally, it would be beneficial for paper's contextualisation to establish some more concrete links between the parameters of PD-SGD and the comparable DP-SGD parameters (i.e. have a simple graph comparing just these two methods as the only 'properly' quantifiable ones showing at which point PD-SGD starts showing better privacy-utility trade-off).

On the empirical results. Firstly, I would really highlight the 'best' results in bold instead of PD-SGD results, as otherwise this is misleading with respect to what the best defence is. It is clear that the method is comparable, but it is also not outperforming all of the baselines, making the abstract a bit misleading with respect to the favourable trade-offs.

Overall I am happy to increase the score should the authors address my concerns (in particular the discussion on fairness and the baseline defence methods).

[1] - Nakai, Tsunato, et al. "SEDMA: Self-Distillation with Model Aggregation for Membership Privacy." Proceedings on Privacy Enhancing Technologies (2024).

**Questions:**

As above: what are the fairness implications of the method? This needs proper evaluation on other datasets (or the same datasets with manual imbalancing).

---

> ### Author Response · Authors · 2024-11-20
> **Responses part 1**
>
> Thank you for your valuable comments and suggestions.
>
> > Firstly, it is not clear what exactly happens if the gradient is implausible - do you remove it or convert all values to 0's or something else entirely? Because 'removal' in a normal sense of a word would result in different sampling for the rest of the users and would cost them more epsilon, making some of them more vulnerable (which has knock-on effects on disparate vulnerability to MIA and how the method affects that).
>
> This is an important point to clarify. Whenever the noisy gradient obtained from the seed batch is implausible, we skip it for model updates for that iteration. However, we do not remove or alter any data samples. The training set and model weights remain unchanged.
>
> In the next learning step, the algorithm reshuffles the training data and so it is free to select a seed batch that includes any data sample. This means that our algorithm does not prevent any sample from having an influence on the final model parameters, unless that sample itself always causes implausible gradients (which if not prevented would lead to privacy leakage).
>
> To clarify this, we conducted a new experiment in which we record the successful update counts for each data sample in the training set and show the distribution in Figure 4 in the appendix of the revised pdf. We can observe that although there is some variation between samples (mean is 107.53 and std is 10.27), no data sample is systematically excluded (min count is 72) when training with PD-SGD.
>
> > Secondly, this has an effect on the fairness of the algorithm: the benchmarks you used only consider the standard (somewhat easy and mostly balanced) datasets. In order to see how the method performs with respect to privacy attacks and subgroup utility, more analysis needs to be done and I request that the authors add a section on how performance imbalance is affected under PD-SGD.
>
> While our PD-SGD method does not discard any data points and ensures all have comparable sampling rates — thus not being designed to have side effects on fairness — we acknowledge that evaluating performance imbalance and subgroup utility is important for understanding how the method performs under privacy attacks. As our paper introduces PD-SGD for the first time, we consider a thorough analysis of its fairness implications to be a valuable direction for future research but outside of scope of this work. We will include a discussion of these considerations in our paper and plan to explore them in depth in subsequent work.
>
>
> > There is also an issue with the way the paper is positioned. I admit I had to re-read the work because it is not immediately clear this method is, in fact, not designed to be differentially private (primarily because of the discussion on adding the gaussian noise and the privacy guarantees afterwards). Just to make it more clear to the reader, I would suggest you point this out early in the text. Especially since following claim only gets introduced in 2.4: 'With some additional randomisation this can be made epsilon-delta DP' What is this additional randomisation and where does it come from? Where is it discussed and how do you convert the guarantees? This sounds (in crude terms) a little bit like k-anonymity here, where you have a rejection sampling based on the number of rows that could have been used to produce the synthetic row (i.e. a probabilistic notion of k). Could you elaborate on why this is/isnt the case and what exactly needs to be done to establish the mapping between the methods?
>
> We apologize for any confusion. To clarify we purposefully focused our efforts away from connecting PD-SGD to DP guarantees, and we will revise the text to make this clear early on in the paper.  The mention of additional randomness in Section 2.4 refers to prior work by Bindschaedler et al. (2017) who add noise to the threshold of their test. In our case, we add Gaussian noise to the gradients but do not add noise to the threshold. That said, exploring how we may adapt PD-SGD to achieve differential privacy is indeed an interesting avenue for future work.

---

> ### Author Response · Authors · 2024-11-20
> **Responses part 2**
>
> > You seem to have picked Selena, which makes sense, but follow-up works have been proposed which significantly improved the effectiveness of the method [1] since this method is empirical, I wonder why you did not evaluate PATE, which actually does come with privacy guarantees and is based on a similar method? If you are only doing training-based methods, then DKFD could be a good one to consider? AdvReg also does not seem like a strong candidate to compare against since it was already shown (in the original paper) to be mostly effective for data with a small number of classes (i.e. not cifar-100), making some of the comparisons less meaningful.
>
>
> Our primary objective in comparing our method with empirical defense mechanisms like SELENA and AdvReg is to demonstrate that our method can achieve comparable performance in terms of the privacy-utility trade-off while also providing formal privacy guarantees that these empirical methods lack. (The formal guarantee here is the rejection of implausible gradients which provably protects any sample that induces large changes in L2-norm with respect to its batch gradient as explained in Section 4.2.)
>
> We are not asserting that our method universally outperforms all others on any given dimension (i.e., privacy *or* utility). Rather, our goal is to highlight its ability to balance privacy and utility effectively.
>
> Regarding your suggestion to evaluate PATE, we acknowledge that it offers strong privacy guarantees. However, PATE requires access to a substantial amount of unlabeled public data from a similar domain as the private dataset. This requirement introduces a different assumption compared to our method and the other baselines we considered. Therefore, we focused on methods that operate under similar conditions to ours, without the need for additional public data.
>
> As for DKFD and the follow-up works to SELENA that have shown significant improvements, we found that they are conceptually similar to SELENA in that they employ distillation techniques to mitigate sensitive knowledge leakage. Additionally, some of these recent works are not open-source, which poses challenges for reproducibility. Moreover, there is recent evidence that DKFD exhibits a worse privacy-utility trade-off than SELENA in certain scenarios (see https://arxiv.org/pdf/2404.17399). That said, we decided we are running the experiments to evaluate DKFD and will update the results when the experiments are finished.
>
> We included AdvReg in our comparisons because it has been used as a baseline in prior works like SELENA and represents a different category of empirical defense mechanisms. Including it allows us to provide a broader perspective on how our method performs relative to various defense strategies.
>
> > it would be beneficial for paper's contextualisation to establish some more concrete links between the parameters of PD-SGD and the comparable DP-SGD parameters (i.e. have a simple graph comparing just these two methods as the only 'properly' quantifiable ones showing at which point PD-SGD starts showing better privacy-utility trade-off).
>
> It is difficult to compare the parameters of PD-SGD with DP-SGD directly because these two methods hold different assumptions and the meanings of their parameters are quite different. For example, $\sigma$ in PD-SGD and DP-SGD are analogous but they are different scales — $\sigma$ of DP-SGD is scaled with batch size while $\sigma$ of PD-SGD is not.  $\gamma$ and $T$ in PD-SGD serve a similar function to the clipping threshold of DP-SGD because they are both used to bound the influence of "outlier samples", they are not one-to-one mappings — and clipping is a deterministic operation whereas our privacy test is probabilistic.
>
> Stepping back, PD and DP are aiming to protect privacy at a different level and in different ways (although they both protect membership privacy) so it is not clear how to directly map their parameters in a meaningful way. But we can compare the two techniques from an empirical privacy perspective as we shown in Figure 2.
>
>
> > It is clear that the method is comparable, but it is also not outperforming all of the baselines, making the abstract a bit misleading with respect to the favourable trade-offs.
>
> The particular phrasing in the abstract was meant to refer to observations such as the one in Figure 2. However, we agree with the reviewer that this could be somewhat misleading and therefore we will soften our claims in the revised version.

---

> > ### Comment · Reviewer_cMXB · 2024-11-21
> > **Response to the rebuttal**
> >
> > I would like to thank the authors for their timely and comprehensive response. My concerns have been addressed and I have raised the score accordingly.

---

### Official Review · Reviewer_usNF · 2024-11-01

**Soundness:** 2
**Presentation:** 4
**Contribution:** 3
**Rating:** 6
**Confidence:** 4

**Summary:**

This paper introduces Plausibly Deniable SGD updates. The key concept involves randomly selecting a seed batch, and if the current update deviates significantly from this seed batch, the update is skipped. To measure the difference between the gradients of the seed batch and the current update, Gaussian noise is added to both, and the log-pdf is bounded by a parameter $\gamma$ (Eq 2). This process is repeated $m$ times for each update, and the update is considered safe if it passes the test at least $T$ times.

**Strengths:**

1. The concept of plausibly deniable updates is both intuitive and innovative.
2. The authors conduct experiments across multiple datasets, showing that this method can provide a favorable privacy-utility tradeoff compared to DP-SGD in some settings. The privacy protection is evaluated using empirical membership inference attacks.
3. The paper is clearly written and easy to understand.

**Weaknesses:**

1. The main assumption behind plausibly deniable updates is that there is no privacy leakage when an update closely resembles updates from other batches. While this assumption is intuitive, it may not hold uniformly across all scenarios.

For example, consider the training of large models with very large batch sizes, where individual sample contributions are averaged out. In this case, even tiny noise may cause the gradients of two large batches to appear deniable, but not necessarily prevent individual samples from being memorized.

Follow-up questions:

(1) How does the rate of deniability scale with batch size across different datasets?

(2) What impact do varying batch sizes have on the overall privacy guarantees?

2. In Table 2, the CIFAR-10 and CIFAR-100 experiments use only 500 and 1000 training samples, respectively. While the performance of plausibly deniable updates in these settings is notable, these sample sizes are too small for a fair comparison with DP-SGD.

Follow-up question:

How do plausibly deniable updates compare with DP-SGD when scaling the training set size for CIFAR-10/CIFAR-100?

3. As discussed in Appendix F, the privacy guarantee for plausibly deniable updates applies on a per-update basis, and the composition of these guarantees over an entire training run remains unclear.

Additional Comments:

1. The intermediate results of training with plausibly deniable updates must remain private, as revealing whether an update was skipped provides side information to the adversary. This limitation may restrict the applicability of the approach in certain scenarios, such as federated learning.
2. When comparing the empirical privacy protection of plausibly deniable updates with DP-SGD, even if a fixed privacy parameter is used for DP-SGD, the clipping threshold still affects the robustness against membership inference attacks. It could be beneficial to tune the clipping threshold to identify the optimal empirical protection for DP-SGD.

**Questions:**

Please see "Weaknesses".

---

> ### Author Response · Authors · 2024-11-20
> **Responses part 1**
>
> Thank you for your valuable comments and suggestions.
>
> > The main assumption behind plausibly deniable updates is that there is no privacy leakage when an update closely resembles updates from other batches. While this assumption is intuitive, it may not hold uniformly across all scenarios. For example, consider the training of large models with very large batch sizes, where individual sample contributions are averaged out. In this case, even tiny noise may cause the gradients of two large batches to appear deniable, but not necessarily prevent individual samples from being memorized.
>
> How does the rate of deniability scale with batch size across different datasets?
> What impact do varying batch sizes have on the overall privacy guarantees?
>
> Regarding the role that batch size plays in terms of privacy, there are extreme edge cases that are unrealistic where the batch size is the entire training set or the batch size is a single example. For more realistic batch sizes there are several tradeoffs and ultimately the behavior depends also on the chosen privacy parameters.
>
> Our findings indicate that as batch size increases, the rate of deniability typically decreases—larger batches more easily pass the privacy test (for fixed privacy parameters) due to the averaging effect you described across different datasets. However, this does not necessarily translate into better privacy protection, as the potential for individual sample contributions to still be inferred remains.
>
> **Impact of batch size on Purchase-100**
>
> | Batch size | Test Acc | Reject Rate | Best Attack |
> |------------|----------|-------------|-------------|
> | 1024 | 0% | 100% | 0.5 |
> | 2048 | 60.10% | 88.94% | 0.62 |
> | 3072 | 64.76% | 10.41% | 0.73 |
> | 4096 | 64.80% | 9.06% | 0.74 |
> | 5120 | 64.73% | 0% | 0.77 |
>
> **Impact of batch size on CIFAR-10**
>
> | Batch size | Test Acc | Reject Rate | Best Attack |
> |------------|----------|-------------|-------------|
> | 1024 | 60.24% | 55.85% | 0.51 |
> | 2048 | 74.35% | 37.63% | 0.51 |
> | 3072 | 80.40% | 20.07% | 0.53 |
> | 4096 | 80.59% | 14.07% | 0.53 |
> | 5120 | 81.57% | 7.41% | 0.54 |
>
>
> Moreover, we found that adjusting other parameters—e.g., $\sigma$, $\gamma$, and threshold can help mitigate these effects, maintaining a balance between utility and privacy across varying batch sizes. For example, for the batch size = 1024, if we double the $\gamma$, we can decrease the reject rate to 56.98% and achieve a test accuracy of 63.87% with Best Attack AUC of 0.68.
>
> These results underscore the importance of carefully tuning all parameters in relation to batch size to uphold robust privacy guarantees while preserving utility.
>
> > In Table 2, the CIFAR-10 and CIFAR-100 experiments use only 500 and 1000 training samples, respectively. While the performance of plausibly deniable updates in these settings is notable, these sample sizes are too small for a fair comparison with DP-SGD.
> How do plausibly deniable updates compare with DP-SGD when scaling the training set size for CIFAR-10/CIFAR-100?
>
> We used small training set sizes for these experiments to ensure the resulting models would be vulnerable to MIA so that it would be clear if the desired level of protection was indeed achieved. However, we also included other experiments in our paper where we use much larger training set sizes (e.g., Table 3). In addition, we conducted further experiments in response to your question using a larger subset of CIFAR-100.  We follow the experiment setting in RMIA which trains a WRN-28-2 from scratch on 25k samples of CIFAR-100.  We show the results in the table below. It can be observed that PD-SGD can successfully defend different MIA attacks for example Attack AUC is decreased significantly from around 81% to 54% by using param setting 2 of PD-SGD. Compared to DP-SGD, PD-SGD provides much better utility.
>
> **Evaluate PD-SGD on CIFAR-100**
> | Method            | Test Acc | P-Attack | R-Attack | S-Attack | C-Attack |
> |-------------------|----------|----------|----------|----------|----------|
> | Non-Private       | 56.27%   | 81.71%   | 81.91%   | 81.85%   | 0.37%    |
> | PD-SGD(param 1)   | 53.63%   | 58.80%   | 52.81%   | 57.46%   | 0.15%    |
> | PD-SGD(param 2)   | 47.07%   | 54.27%   | 50.56%   | 50.00%   | 0.12%    |
> | DP-SGD(eps=8)     | 18.24%   | 52.29%   | 49.58%   | 51.03%   | 0.11%    |
>
>
> [RMIA] Zarifzadeh, Sajjad, Philippe Liu, and Reza Shokri. "Low-Cost High-Power Membership Inference Attacks." Forty-first International Conference on Machine Learning. 2024.

---

> ### Author Response · Authors · 2024-11-20
> **Responses part 2**
>
> > As discussed in Appendix F, the privacy guarantee for plausibly deniable updates applies on a per-update basis, and the composition of these guarantees over an entire training run remains unclear.
>
> That’s a fair point. Our proposed method is based on privacy testing per update. The composition of all the updates is interesting and we leave formal investigation of this part for future work.
>
>
> > The intermediate results of training with plausibly deniable updates must remain private, as revealing whether an update was skipped provides side information to the adversary. This limitation may restrict the applicability of the approach in certain scenarios, such as federated learning.
>
> The reviewer is correct. Our solution assumes that no model intermediate updates are accessible to an adversary. While such assumptions may prevent our solution from being used in a setting such as Federated Learning, we intend our technique to mostly be used in a centralized learning environment where adversaries only observe the final model weights (or run inference with the trained model as a black-box). Note that this is a similar setting as hidden state differential privacy analyses [1-3] that focus on the privacy of the final model parameters while assuming the confidentiality of the training dynamics. We will further emphasize this fact in the final version.
>
> [1] Ye, Jiayuan, and Reza Shokri. "Differentially private learning needs hidden state (or much faster convergence)." Advances in Neural Information Processing Systems 35 (2022): 703-715.
>
> [2] Altschuler, Jason, and Kunal Talwar. "Privacy of noisy stochastic gradient descent: More iterations without more privacy loss." Advances in Neural Information Processing Systems 35 (2022): 3788-3800.
>
> [3] Chen, Ding, and Chen Liu. "Differentially Private Neural Network Training under Hidden State Assumption." arXiv preprint arXiv:2407.08233 (2024).
>
>
> > When comparing the empirical privacy protection of plausibly deniable updates with DP-SGD, even if a fixed privacy parameter is used for DP-SGD, the clipping threshold still affects the robustness against membership inference attacks. It could be beneficial to tune the clipping threshold to identify the optimal empirical protection for DP-SGD.
>
> Following your suggestion, we have fixed all other parameters and tuned the clip threshold for DP-SGD as shown in the following table. We can observe that even though the clip threshold changes, the model’s utility and privacy are almost the same. However, during these experiments, we do find that if the clip threshold is changed, the learning rate also needs to be tuned properly to get the optimal utility. It makes sense that the impact on privacy of the clipping threshold should not be substantial since in DP-SGD the noise added to the gradient is scaled by the clipping norm.
>
> **Impact of Clip threshold of DP-SGD**
> | Clip threshold | Test Acc | P-Attack | R-Attack | S-Attack | C-Attack |
> |----------------|----------|----------|----------|----------|----------|
> | 0.1            | 93.49%   | 0.54     | 0.56     | 0.54     | 0.18%    |
> | 1              | 93.56%   | 0.54     | 0.56     | 0.54     | 0.18%    |
> | 10             | 93.54%   | 0.54     | 0.57     | 0.54     | 0.20%    |

---

> > ### Comment · Reviewer_usNF · 2024-11-26
> > **Discussion**
> >
> > Thank you for your response and the new experiments. I have two follow-up comments:
> >
> > 1. Regarding the impact of batch size
> >
> > The new results are interesting. I believe the finding that batch size affects the empirical privacy guarantee, even though the privacy setup (parameters) for plausibly deniable privacy does not change, is significant. **Please upload a revision to prominently include these results.**
> >
> > 2. Regarding the number of training samples in Table 2
> >
> > My question pertains to Table 2, where the models are pre-trained ViTs, while the new experiments involve training from scratch. Could you add experiments that use a larger number of training samples in the fine-tuning setting? DP-SGD is known to require a large training set, and the current setup for fine-tuning only uses a few hundred samples, despite the availability of a larger training set.
> >
> > I agree that the current results suggest PD-SGD outperforms DP-SGD in few-shot settings, but it would still be valuable for the community to see fine-tuning results with a larger training set.

---

> > > ### Author Response · Authors · 2024-11-27
> > >
> > > Thank you for your response.
> > >
> > > > Regarding the impact of batch size
> > >
> > > We have added these results to Appendix D.3. We appreciate the reviewer for raising these valuable points regarding batch size, as they have significantly helped us improve our paper
> > >
> > > > Regarding the number of training samples in Table 2
> > >
> > > We appreciate the reviewer's concern regarding our choice of using a small subset of CIFAR-10 and CIFAR-100 for fine-tuning the ViT model. We would like to clarify the reasoning behind this decision.
> > >
> > > In the initial phase of our experiments, we employed larger subsets of CIFAR-10 and CIFAR-100 for fine-tuning. However, we observed that due to the strong generalization capability of the pre-trained ViT model, the attack AUC was close to 50%, indicating near-random performance. This made it challenging to demonstrate the effectiveness of different defense mechanisms, as the attacks were not successful enough to highlight significant differences between them.
> > >
> > > To effectively showcase the impact of our defense mechanism, we reduced the number of training samples. This adjustment increased the attack success rate, allowing us to more clearly compare the performance of various defense strategies.
> > >
> > > Furthermore, to demonstrate that our proposed method is nevertheless applicable to larger datasets, we conducted experiments using substantial numbers of training samples: 25,000 for Purchase-100, 30,000 for CIFAR-10 trained from scratch, and 25,000 for CIFAR-100 trained from scratch.
> > >
> > > To directly address the reviewer's concern, we have decided to run additional experiments using a larger training set on CIFAR-100 for fine-tuning. We will report these results once the experiments are completed.

---

> > > > ### Author Response · Authors · 2024-12-02
> > > > **Update on additional experiments: larger training set on CIFAR-100**
> > > >
> > > > We have finetuned ViT model with a larger subset of CIFAR-100 i.e., using 10K for training and 10K for testing and rest of data for shadow datasets. We report results in the following table:
> > > >
> > > > **Evaluate PD-SGD on large subset of CIFAR-100**
> > > >
> > > > | Method       | Test Acc | P-Attack | R-Attack | S-Attack | C-Attack |
> > > > |--------------|----------|----------|----------|----------|----------|
> > > > | Non-Private  | 82.94%   | 0.56     | 0.57     | 0.56     | 0.19%    |
> > > > | PD-SGD(param 1) | 80.29%   | 0.52     | 0.52     | 0.51     | 0.11%    |
> > > > | PD-SGD(param 2) | 78.25%   | 0.51     | 0.51     | 0.51     | 0.08%    |
> > > > | DP-SGD(eps=8) | 77.13%   | 0.52     | 0.54     | 0.52     | 0.13%    |
> > > >
> > > > We can observe that without any defense, the attack AUC is around 0.56, while with PD-SGD, the attack AUC decreases to 0.52-0.51. For utility, PD-SGD can achieve 80.29% test accuracy while DP-SGD with eps=8 can only achieve 77.13% test accuracy.

---

### Official Review · Reviewer_7N4j · 2024-11-03

**Soundness:** 2
**Presentation:** 2
**Contribution:** 2
**Rating:** 5
**Confidence:** 3

**Summary:**

The authors propose an alternative to empirical MIA defenses and DP-SGD and claims higher utility while making compromises in the theoretical guarantees in comparison to DP but in comparison to other works it still provides some. The authors provide both theoretical as well as empirical evidence that their method can be an alternative.

**Strengths:**

- providing a new method closing the gap between purely empirical defenses and DP-SGD by introducing a new method that has theoretical guarantees that are weaker than with DP but at the same time providing higher utility.
- theoretical results regarding the new method
- empirical results and ablation studies comparing the method to other alternatives

**Weaknesses:**

I cannot recommend the acceptance of the paper at the moment as especially the empirical part is not strong enough to understand the contribution and the advantage of the method. I am willing to increase my score when the below points and questions are addressed or there are some contributions or points that I misunderstood.

- **W1: Comparison to other defenses without any error bars.** The authors compare in Section 5 to other defenses using multiple attacks. Unfortunately, there are not error bars for test accuracy or the membership inference attack metrics. Related prior work on few-shot image-classification using fine-tuning (so exactly the same setting that the authors are operating in) by Tobaben et al. [1] found that there is quite a difference in test accuracy over multiple repeats (see e.g., Figure 1 of [1]). Other work focusing on membership inference attacks like the recent RMIA [2] use repeats for comparisons like in Table 3 of [2]. In my opinion, the proposed method performs not as clearly better to justify the bold claim in the contributions: *"Results demonstrate that PD-SGD offers a superior privacy-utility trade-off compared to alternatives."*. E.g., Table 2: DP-SGD on CIFAR-10 ($\epsilon=8$) is nearly as good in test accuracy and comparable in the resistance to attacks. SELENA is comparable in test accuracy and resistance to attacks throughout all experiments in Table 2 and 3. Note that I am not doubting the claims of the authors entirely but I feel that without error bars the conclusions made by the authors are too strong. The authors could improve here by providing error bars.

- **W2: Runtime comparison.** One concern I have with the method is the runtime of it. The authors write in Section 3.3: *"In experiments (supplementary materials) we find that although PD-SGD is slower than
SGD, it is often much faster than DP-SGD for a single training step, in large part because it does not require calculating per-example gradients."* Table 6 compares the run time for the same amount of data and find that DP-SGD is around $38 \times$ and PD-SGD is around $15.7 \times$ slower than SGD for the same amount of data. When comparing to prior work by Bu et al. [3] and Beltran et al. [4] that the authors cite in the introduction the number for DP-SGD seems very high (Figure 3 of [3] suggests more factor 3 with opacus and [4] shows for the same model architecture as the authors use a factor of $2.8 \times$). Furthermore, both related works discuss methods that make DP training faster. The authors should discuss mention these methods (e.g., GhostClipping) and validate their runtime measurements as the gap seems strikingly large.

- **W3: Experimental Setting is not described in detail:** In is unclear from the main text how many data points are used in the experiments in Section 5 (this is a very important information), it is unclear from the paper how shadow datasets are used (e.g., for S-Attack and C-Attack), how many shadow models are trained and if the attacks are run in offline or online setting (e.g., at least C-Attack). It is unclear what the hyperparameters for all other defenses than PD-SGD are and how they were selected. Furthermore, it is not clear from the main text what subsets of the models were fine-tuned. The authors could improve their paper by providing all details to ensure reproducibility of the paper.

Minor:
- I looked at the `ml_privacy_meter` at https://github.com/privacytrustlab/ml_privacy_meter that you mentioned but I think it doesn't implement LiRA in its current form. Browsing through the commits I found that perhaps the package changed significantly in the meantime. I would recommend the authors to refer to a particular git hash and include that link to make sure that the reproducibility is ensured.

- In Line 363: There are typos with the dataset names.

- I discussed Tobaben et al. [1] in the upper part as it is a very related work on few-shot fine-tuning but it is not mentioned in the paper. I recommend the authors to include it in their discussion as a related work.

[1] Tobaben, M., Shysheya, A., Bronskill, J. F., Paverd, A., Tople, S., Zanella-Beguelin, S., ... & Honkela, A. On the Efficacy of Differentially Private Few-shot Image Classification. TMLR 2023.

[2] Zarifzadeh, S., Liu, P., & Shokri, R. (2024). Low-Cost High-Power Membership Inference Attacks. In ICML 2024.

[3] Bu, Z., Mao, J., & Xu, S. (2022). Scalable and efficient training of large convolutional neural networks with differential privacy. in NeurIPS 2022.

[4] Beltran, S. R., Tobaben, M., Loppi, N., & Honkela, A. (2024). Towards Efficient and Scalable Training of Differentially Private Deep Learning. arXiv:2406.17298.

**Questions:**

- Q1: Why is DP-SGD so slow in your setting? I assume you implement it using off-the-shelf opacus? (Re W2)
- Q2: How are the attacks run? E.g., how many shadow models? (Re: W3)
- Q3: What are the hyperparameters for the other methods and how are they determined? (Re. W3)
- Q4: In line 158 you say that plausible deniability can be made DP. Is this applicable to your method as well?
- Q5: Could you provide some intuition on how to pick the hyperparameters in practice for your method? DP hyperparameters can be connected to MIA success bounds (e.g., Kairouz et al. [5]). How would it be for your method?
- Q6: Could you provide access to your code please? I would be interested in reviewing the details of that implementation.

[5] Kairouz, P., Oh, S., & Viswanath, P. (2015, June). The composition theorem for differential privacy. In International conference on machine learning (pp. 1376-1385). PMLR.

---

> ### Author Response · Authors · 2024-11-20
> **Responses part 1**
>
> Thank you for your valuable comments and suggestions.
>
> > Comparison to other defenses without any error bars
>
> We agree that this is important in demonstrating the robustness of our results through statistical measures such as error bars, particularly when comparing our PD-SGD method against other defenses across multiple metrics. We have started additional experiments to calculate standard deviations for all reported metrics.
>
> Please see the revised version, where we have already incorporated error bars for the test accuracy and membership inference attack metrics for CIFAR-10 and CIFAR-100, as shown in the updated Tables 2. Our conclusions remain the same. Experiments for other datasets and settings are still running. But we will ensure that all results throughout the paper include this level of detail. This additional data will allow us to more precisely assess the performance of PD-SGD relative to other methods like DP-SGD and SELENA.
>
> We also appreciate your references to related literature and we will cite them.
>
> > Runtime Comparison
>
> This is a critical point regarding runtime comparisons. In our experience runtime can vary substantially based on several factors including model architecture specifics, training settings, and computational environment (such as CUDA versions and the specifics of the hardware used). As stated in the paper, we are using off-the-shelf Opacus. We believe differences in training setting account for discrepancies between our results and other papers' results. (For example, we are only finetuning the last layer of Vit model and [4] are finetuning the whole model; we are using open clip vit-b-16 model while [3] use a different version of Vit model from timm.)
>
> We decided to conduct further measurements to investigate this. We measured the running time for the WRN-16-4 model and trained it from scratch with batch size=1024. We find that even though PD-SGD is slower than SGD, it is still faster than DP-SGD. And the factor between DP-SGD and SGD also changes to around 7X due to a different model architecture and training from scratch setting.
>
> **Computational Time per step for WRN-16-4 on CIFAR-10**
> | Method | Time (ms)          |
> |--------|--------------------|
> | DP-SGD | 2492.11 ± 8.06     |
> | PD-SGD | 1780.16 ± 15.72    |
> | SGD    | 344.47 ± 0.20      |
>
> Regarding works that attempt to speedup DP-SGD, we appreciate the pointer and are happy to cite them. We can include a comparison if the reviewer feels it is warranted. However, we emphasize that our proposed method avoids the main bottleneck of every implementation of DP-SGD, which is the necessity to store individual gradients in memory.
> Also, on top of avoiding storing individual gradients in memory and thus improving runtime, the main contribution of our work is to propose a novel approach for obtaining privacy for deep learning, yielding a favorable privacy utility trade-off. An exact characterization of the runtime benefits of our privacy notion is left as future work.

---

> ### Author Response · Authors · 2024-11-20
> **Responses part 2**
>
> > Experimental Setting is not described in detail
>
> Due to the page limit, we list details of datasets, models and our experiments setup in Appendix C.  Following your suggestion, we added more details about the attacks and defense mechanisms which is highlighted in the revised pdf Appendix C.3.
>
> > ml_privacy_meter change
>
> We acknowledge that the ml_privacy_meter codebase has undergone significant changes since our submission. To ensure the exact replication of our experiments, we are providing the specific git commit hash of the version we used. It can be accessed directly at [ml_privacy_meter at commit 173d4ad](https://github.com/privacytrustlab/ml_privacy_meter/tree/173d4ad80f183ae6e1867b2793dfffe0633107d0). Also, here is a direct link to the [main.py](https://github.com/privacytrustlab/ml_privacy_meter/blob/173d4ad80f183ae6e1867b2793dfffe0633107d0/benchmark/main.py) file where the LiRA attack implementation as reference_in_out_logit_pdf_fixed can be reviewed as used in our evaluations. More description about the method can be found [here](https://github.com/privacytrustlab/ml_privacy_meter/tree/173d4ad80f183ae6e1867b2793dfffe0633107d0/benchmark).  We hope this addresses your concerns.
>
> > In line 158 you say that plausible deniability can be made DP. Is this applicable to your method as well?
>
> This specific statement is about the method proposed by Bindschaedler et al. (2017) in the context of synthetic datasets and described in Section 2.4 as part of the background. In our setting, we believe similar reasoning may apply, but we intentionally steered away from drawing a formal connection of our method to differential privacy (which we leave for future work).  Also note that our technique does not randomize the threshold. Our focus in this paper is demonstrating the potential of our technique both through its analytical connection to L2 norm differences between batches' gradients and as an empirical defense against membership inference attacks achieving a better utility vs privacy trade-off compared to DP.
>
>
>
> > Could you provide some intuition on how to pick the hyperparameters in practice for your method? DP hyperparameters can be connected to MIA success bounds (e.g., Kairouz et al. [5]). How would it be for your method?
>
> Thank you. We have expanded our discussion of privacy parameter tuning for our PD-SGD method. There is a theoretically-guided method for hyperparameter tuning discussed in Section 4.2. However, in practice we recommend as an alternative to tune the noise scale $\sigma$ first to obtain acceptable utility and then tune $\gamma$ and $T$ to achieve the desired privacy level.
>
> Regarding the connection of DP hyperparameters to MIA success bounds, we are not sure what the reviewer means. Could you expand on that point?
>
>
> > Could you provide access to your code please? I would be interested in reviewing the details of that implementation.
>
> We are providing a link to an anonymous GitHub to be shared only to reviewers due to institutional restrictions at this stage.

---

> > ### Comment · Reviewer_7N4j · 2024-11-25
> >
> > Apologies for replying so late, I tried my best to reply ASAP.
> >
> > **Regarding your question for clarification**
> > > Regarding the connection of DP hyperparameters to MIA success bounds, we are not sure what the reviewer means. Could you expand on that point?
> >
> > I am mostly asked if there is some better way than heuristics of picking the hyper-parameters, e.g., (epsilon,delta)-DP can be connected to MI-Attack upper bounds (e.g., like in the Kairouz paper that I cited).
> >
> > I was wondering how to tune it in practice. You wrote: "We have expanded our discussion of privacy parameter tuning for our PD-SGD method". Where can I find this?
> >
> > **General response**
> >
> > Thanks for the responses to the questions and for partially addressing the weaknesses. You addressed partially W1 (some error bars) and made some important clarification regarding W2 (getting numbers more in line with other literature here), W3 some important experiment settings.
> >
> > Unfortunately, you don't seem to address the following things in your revision:
> > - errorbars for all experiments (W1)
> > - new runtime numbers are not included in the revised version and you don't discuss them really well, I think it is a reasonable argument that you make that the runtime is not the main selling point but in Table 6 you write: "We can observe that although PD-SGD is a little bit slower than SGD, it takes less time than DP-SGD." while the actual numbers suggest DP-SGD is around $38 \times$ and PD-SGD is around $18 \times$ slower than SGD. I do not follow why $18 \times$ slower is "a little bit slower".
> > - you don't mention that your numbers are few-shot numbers in the main text and that you are fine-tuning in many experiments (W3)
> > - You wrote "We also appreciate your references to related literature and we will cite them.", but I don't see any added references
> >
> > I feel like these points are necessary and you somewhat addressed partially, but I would feel it more convincing if you actually did this in a revision rather than promising something. ICLR gives you the opportunity to do that.
> >
> > I will raise my score if you revise the version as you promise but I don't see where you did certain revisions. I am not sure if you uploaded actually the revision with all changes or if some revised things are not highlighted.

---

> > > ### Author Response · Authors · 2024-11-25
> > >
> > > We thank the reviewer for the detailed responses. We apologize for the miscommunication and are sorry that we did not upload the full-revised version of pdf at that time. We were waiting for all experiments to finish to include the results for error bars of Table 2 and 3. Now experiments are finished and we uploaded the new version of pdf for review. We will respond to your concern one by one.
> > > > Regarding your question for clarification
> > >
> > > For the upper bound: Thank you for clarifying. Yes, this is an interesting point. Informally, the quantity $q(d)$ which bounds the probability of passing the test (Section 4.2) should yield a bound on MIA success rates. Therefore, a strategy to tune privacy hyperparameters is to find a combination of $\gamma$ and $\sigma$ that provides acceptable utility while keeping $q(d)$ small enough.
> > >
> > >
> > > For hyperparameters tuning in practice:  We have included a detail discussion in appendix E and for your convenience, we also state it here:
> > > **Theory-based strategy:**
> > > As explained in Section 3.1, by tuning $\sigma$ and $\gamma$, we can make $q(d)$ arbitrarily small. If we have a desired bound on $d$, then we can find combinations of $\sigma$ and $\gamma$ that achieve the desired effects (e.g., see Figure 1. This can for example be done through a grid search.
> > >
> > > **Empirical strategy:**
> > > Alternatively, we found that the following two-step strategy is easy to follow and yields good trade-offs.
> > > Step 1: tune the noise $\sigma$ to achieve acceptable utility, ignoring the privacy test. This helps determine an upper limit for utility.
> > > Step 2: tune $\gamma$ and the threshold $T$, which allows for fine-grained control over the privacy-utility trade-off. We used this two-step sequential tuning approach in our experiments.
> > >
> > > A useful heuristic while tuning $\gamma$ and $T$ is to monitor the rejection rate. However, note that there exists favorable trade-offs for a wide-range of rejection rates, and a useful rule of thumb is therefore only to avoid extreme values (e.g., 0% --- no privacy guarantee; 100% --- no utility / full privacy).
> > >
> > > > errorbars for all experiments (W1)
> > >
> > > We have included all error bars for Table 2 and 3. Our conclusions remain the same.
> > >
> > > > new runtime numbers are not included in the revised version
> > >
> > > We have included it in the new version of pdf in appendix D.1 and update the table 6. We also modified the claim as you suggested. In these experiments, PD-SGD is significantly slower than SGD but also substantially faster than DP-SGD.
> > >
> > >
> > > > don't mention that your numbers are few-shot numbers
> > >
> > > Following your suggestion, we have made it clearer in experimental setup in section 5.1. Please see the highlighted part in the pdf.
> > >
> > > > Regarding references
> > >
> > > We have expanded our related work section (section 2.3) and experimental setup (section 5.1) to discuss the added references. Please see the highlighted part in the pdf.

---

> > > > ### Comment · Reviewer_7N4j · 2024-11-26
> > > >
> > > > Thanks for the responses and for revising what you promised earlier. You improved significantly the description and experiments of the experiments. I will raise my score accordingly.

---

### Official Review · Reviewer_9rjs · 2024-11-04

**Soundness:** 2
**Presentation:** 3
**Contribution:** 2
**Rating:** 5
**Confidence:** 3

**Summary:**

The paper proposes a new notion of privacy tailored to noisy SGD, called plausible deniability, inspired by prior work on synthetic data. The notion could be restated as follows. Given $k \geq 1$ mini-batch gradients $g_j$ and their noisy versions $\tilde g_j = g_j + \mathcal{N}(0, \sigma^2)$, a gradient update step satisfies $(s, \sigma, T, \gamma)$-plausible deniability if for a given $s \in [k]$, we have $|\\{ \\mathrm{LR}\_{s,i}(\tilde g_s) \leq \gamma \mid i \neq s \\}| \geq T$, where $\\mathrm{LR}\_{s,i}: \mathbb{R}^{d} \rightarrow \mathbb{R}$ is the absolute value of the Neyman-Pearson log-likelihood ratio between two simple hypotheses $H_0: \tilde g_s \sim \mathcal{N}(g_s, \sigma^2 I_d)$ and $H_1: \tilde g_s \sim \mathcal{N}(g_i, \sigma^2 I_d)$:

$ \\mathrm{LR}\_{s,i}(\tilde g_s) = |\log p(\tilde g_s \mid \mathcal{N}(g_s, \sigma^2)) - \log p(\tilde g_s \mid \mathcal{N}(g_i, \sigma^2))|,$
where $p(\cdot \mid P)$ is a pdf of the distribution P.

In other words, a noisy gradient update is plausibly deniable if there exist $T$ other mini-batches that could have produced it aside from the original mini-batch, with a given pre-specified confidence level $\gamma$. The paper proposes PD-SGD, a variant of noisy SGD which rejects gradient updates that do not pass the plausible deniability test. The paper shows that it outperforms empirical defenses in terms of the trade-off between empirical membership inference attack performance and utility.

**Strengths:**

This paper aims to strike a balance between provable privacy guarantees and the loss in utility and computational overhead, by introducing a new notion of privacy inspired by prior work on synthetic data. The notion should significantly improve performance of DP-SGD as it does not require clipping. The approach is novel and interesting.

The paper does a good job at positioning their work within the landscape of prior research. The usage of four attack types provides a fairly solid basis for an empirical evaluation of the proposed method.

**Weaknesses:**

Although I think the approach is quite interesting, I believe in the current version it misses the mark on one of the two stated goals of the paper — providing robust privacy guarantees.

This is because the definition lacks clear operational meaning, which is manifested by the following concrete issues:
   1. Crucially, what does plausible deniability mean for, e.g., membership inference, attribute inference, or reconstruction robustness?
   2. By fixing the Neyman-Pearson statistic threshold $\alpha$/$\gamma$, the definition inadvertently specifies the test error rates as a function of gradient difference $d = ||g_s - g_i||_2^2$. Depending on the noise scale, and the distribution of gradient differences, the test's Type I/Type II error could be significant — and again, the definition hides this fact by fixing $\gamma$ in advance. E.g., if values of $d$ in practice happen to be close enough to the noise scale, the test will have high error rates, meaning that the $T$ plausibly deniable candidates could actually be relatively different from the seed gradient.
   3. The definition concerns itself with plausible deniability of the mini-batch gradients and not individual per-sample gradients. What is the protection for individual samples? In particular, the test boils down to approximately verifying whether there exist other mini-batches within an L2 ball around the seed mini-batch. What if there exist many mini-batches with similar L2 norm, but it is the _direction_ of the gradient that is significantly impacted by a single outlier example in the seed mini-batch? The paper lacks both formal and informal discussion on the level of protection for individuals, and the discussion in Sec 4.2 does not address this sufficiently.
  4. As the parameters are not immediately interpretable, the parameter tuning process lacks clarity. The authors give a rationale for their choices in Section 4.2 and Appendix D.2, but it’s still hard to see how someone would apply this method in real-world situations.

If these issues are not fixed, the method should be treated as primarily empirical defense, and would likely need a broader set of empirical evaluations.

**Questions:**

- In Fig. 2, why define advantage as 2 AUC - 1 if the standard definition is 2 balanced_accuracy - 1 = TPR - FPR?
- In Fig. 2, isn't there a parameterization of AdvReg that could be used to produce a curve instead of a single point?

---

> ### Author Response · Authors · 2024-11-20
>
> Thank you for your valuable comments and suggestions.
>
> > what does plausible deniability mean for, e.g., membership inference, attribute inference, or reconstruction robustness?
>
> We define plausible deniability as a new privacy notion in our setting, with the main goal of protecting membership privacy for the training data. However, the notion is in principle more general. It guarantees gradient updates can be plausibly denied in the following sense. Imagine someone comes to the model trainer and accuses them of using a particular data point (x,y) at a specific training iteration. If the model was trained with PD-SGD, the owner can (in principle) readily deny that (x,y) was used (regardless of whether it was in fact used) by showing that pairs of batches from the training data that contain (x,y) and some that do not and showing they both have similar gradients updates to the ones used to update the model parameters. We demonstrate through experiments that such a notion of privacy does offer concrete protection against membership inference attacks.
>
> > if values of d in practice happen to be close enough to the noise scale, the test will have high error rates, meaning that the T plausibly deniable candidates could actually be relatively different from the seed gradient.
>
> If values of d are close to the noise scale then it becomes difficult for adversaries to detect differences between batches given the noise. More generally, our technique like other techniques requires careful parameter tuning to achieve its objectives.
>
> > The definition concerns itself with plausible deniability of the mini-batch gradients and not individual per-sample gradients. What is the protection for individual samples? In particular, the test boils down to approximately verifying whether there exist other mini-batches within an L2 ball around the seed mini-batch. What if there exist many mini-batches with similar L2 norm, but it is the direction of the gradient that is significantly impacted by a single outlier example in the seed mini-batch? The paper lacks both formal and informal discussion on the level of protection for individuals, and the discussion in Sec 4.2 does not address this sufficiently.
>
> Ensuring plausible deniability of the mini-batch gradients does protect individual samples. If there are multiple mini-batches with similar norms but different directions the test will behave as intended, because the condition checking plausible deniability is on the gradients, not their norm. (It is not the norm of the gradients that matters but rather the norm of gradient pair differences. For such norms to be small so that the test passes the two gradients have to point in a similar direction or both have small norms.). We will revise our paper to expand on this discussion, following the reviewer’s recommendations.
>
> > As the parameters are not immediately interpretable, the parameter tuning process lacks clarity. The authors give a rationale for their choices in Section 4.2 and Appendix D.2, but it’s still hard to see how someone would apply this method in real-world situations.
>
> We will expand our discussion of privacy parameter tuning for our PD-SGD method. Section 4.2 discusses theoretical guidelines for parameter tuning. If one has a desired bound on $d$ then the results in that section can be applied with a grid search to find optimal parameter combinations. However, we recognize that there are practical complexities and one may not have a bound on $d$. In practice, we have found that the following strategy is easy to follow and yields good tradeoffs. Step 1: tune the noise scale $\sigma$ to achieve acceptable utility, ignoring the privacy test. This first step helps establish an upper limit for utility. Step 2: tune $\gamma$ and the threshold $T$, which allows for fine control over the privacy-utility tradeoff. This sequential tuning approach was used in our experiments.
>
> > why define advantage as 2 AUC - 1 if the standard definition is 2 balanced_accuracy - 1 = TPR - FPR?
>
> Apologies. Our goal was not to redefine advantage but to express attack success rate in a [0-1] range for ease of comparison. We have revised Figure 2 according to your suggestion.
>
> > isn't there a parameterization of AdvReg that could be used to produce a curve instead of a single point?
>
> Following your suggestion, we added more points for AdvReg to produce a curve in the updated Figure 2.

---

> > ### Comment · Reviewer_9rjs · 2024-11-25
> >
> > Thanks for the response. I have checked the updated text in the Appendix detailing the tuning strategies.
> >
> > I don't think the response and the updates have substantially addressed my concerns. To reiterate what I said in the initial review:
> > >  We define plausible deniability as a new privacy notion in our setting, with the main goal of protecting membership privacy for the training data.
> >
> > My main point is exactly about the definition. This definition hides the level of privacy in terms of test error rate through implicit dependence on the the gradients ($q(d)$). Because of this, the following interpretation:
> >
> > >  If the model was trained with PD-SGD, the owner can (in principle) readily deny that (x,y) was used (regardless of whether it was in fact used)
> >
> > ...is mostly invalid. This is because the owner's level of plausible deniability varies arbitrarily due to the non-trivial interactions between $d$, $\sigma$, $\gamma$. As mentioned in the original review, this challenges the value of the approach in terms of operational meaning and interpretability, making this primarily an empirical defense.
> >
> > > Ensuring plausible deniability of the mini-batch gradients does protect individual samples [...] We will revise our paper to expand on this discussion, following the reviewer’s recommendations.
> >
> > Although I can see that with $q(d)$ being close to zero, this might be true, I do not see a formal argument clearly describing the protection of individual samples in the updated version. I believe the paper this is a necessary component for understanding the privacy protection of the proposed definition.
> >
> > Because of these issues, I would keep my score.

---

> > > ### Author Response · Authors · 2024-11-27
> > >
> > > We are sorry that we have so far been unable to address your concerns. We are not sure if we fully understand your concerns.
> > >
> > > In that respect, could you please clarify whether the following points from our prior responses were acknowledged?
> > >
> > > (1) Do you acknowledge the point about the noise scale in relation to the gradient differences?
> > >
> > > (2) Do you acknowledge the point about the test being based on the norm of *gradient differences*, and not on the *norm of gradients*?
> > >
> > > We believe both points are critical to understanding why the test offers protection.
> > >
> > > Regarding the definition and what it “hides” there are a few points to clarify:
> > >
> > > (a) We parameterized our technique in terms of $\sigma, \gamma, T$ because we felt it was natural, particularly when it comes to domain experts and practitioners who may have an idea of the noise level they want and the tightness of the log ratio for plausibility.
> > >
> > > However, other parameterizations are feasible. For example, it may be reasonable to define $\sigma = c \sqrt{d}$ for some chosen $d$ and some $c > 0$ and have $c$ and $d$ instead as parameters. One could also define $\gamma$ in terms of $d$ and/or $c$.
> > >
> > > We don’t believe the parameterization here is hiding anything or that all else equal it changes the protection offered.  By analogy, DP-SGD could be parameterized with a noise scale based on the clipping bound $C$ instead of $\sigma$ say $\sigma’ = a / C$ for some $a>0$ since the noise added with DP-SGD is of scale $C \sigma$ but that would not change the guarantee. In fact, some papers suggest including the clipping bound into the learning rate, but again this has no effect on the guarantee.
> > >
> > > (b) For PD-SGD once $\sigma, \gamma, T$ are fixed, so is the guarantee; it does not vary. Whether it is sufficient for a particular application is a different question, but that is why parameters have to be set carefully (and this applies to any privacy notion).
> > >
> > > Note that $d$ in our case is not a parameter, and a strength of our technique is that we do not need to know it. Albeit we may set the parameters to protect privacy in the sense of being able to plausibly deny any change to the batch gradient of magnitude $d$ or more. To reiterate: for any $d > 2 \sigma^2 \gamma$ the probability of passing the test, $q(d)$, is bounded and decreases exponentially fast in $d$ as described in Lemma 2.
> > >
> > > (c) We acknowledge that it is difficult to interpret the privacy guarantee directly in terms of the parameters (as other reviewers pointed out). This is a common challenge of many privacy notions. Even for DP, there is literature discussing how to relate $\varepsilon$ to more easily interpretable privacy concerns (c.f. differential identifiability).
> > >
> > > Regarding what our proposed technique guarantees:
> > >
> > > (d) The type I/II error rates of the privacy tests are not meaningful by themselves. We have to consider both the noise added to the gradient and the test. Thinking of the privacy test as a statistical hypothesis test — where one has to tightly control the tradeoff between type I and type II errors — is potentially misleading. We can construct scenarios in which the test error rates can be changed arbitrarily without affecting the privacy guarantee.
> > >
> > > Imagine a hypothetical scenario where we have a "sensitivity" oracle that gives us a bound on the maximum norm of any individual example’s gradient and thus a bound on the largest $d$ caused by adding/removing an example for a given dataset. If the noise scale $\sigma$ is sufficiently large, i.e., $\sigma =c \sqrt{d}$ for some $c > 1$ the noise added would be sufficient to hide any gradient difference caused by adding/removing an example. In that case, PD-SGD would be equivalent to DP-SGD (since no clipping would occur anyway) except that the privacy test could reject the noisy gradient update. Since the test's probability of passing depends also on $\gamma$ and $T$ the test could be made arbitrarily easy/difficult to pass. But privacy is protected (in the DP sense) no matter what happens with the test.
> > >
> > > We need the privacy test exactly for the case where the noise is not sufficiently large. Suppose $\sigma \ll \sqrt{d}$. In that case, there is the potential for an individual example to have an immense impact on the batch gradient, changing it completely. DP-SGD protects this case by clipping individual example gradients. Our method takes a different approach, gradients aren't clipped but the leakage is prevented due to the privacy test. If there are no other batches that could yield such a noisy gradient, then the test will reject the update with overwhelming probability (as $q(d)$ will be very small). If there are other plausible batches then the influence of the individual example is anomalous with respect to its seed batch gradient but its inclusion in the dataset can be plausibly denied by pointing to these other batches.

---

### Author Response · Authors · 2024-11-20
**General Response:**

Thank you for your insightful comments and feedback. We will post an individual response to each review. We wanted to point out that we just posted a revised PDF with the following changes.

(1) Added error bars to Table 2 (reviewer 7N4j) and removed bolding from Tables 2 and 3 (reviewer cMXB)

(2) Updated Figure 2 to define advantage in terms of the balanced accuracy (reviewer 9rjs) and show more points for the tradeoff for AdvReg (reviewer 9rjs).

(3) Added more details on experimental setup for attacks and defenses in Appendix C.3 (reviewer 7N4j)

(4) Added a new experiment (Appendix D.3 and Figure 4) to show the distribution of training set examples being used for updating parameters (reviewer cMXB).

Also, some of the experiments we started due to reviewers' suggestions are still running. But we will update our response once we get those results.

---

### Author Response · Authors · 2024-11-25
**General Response 2:**

We thank all reviewers again for their valuable feedback and insightful suggestions. We have updated the PDF with the following changes:

(1) Added error bars for the rest of the results in Table 2 and Table 3. Now we have error bars for all main results.

(2) Updated Table 6 and added new runtime results for training from scratch on CIFAR-10.

(3) Updated experiments setup to make it clearer.

(4) Added new references to related work and experiments.

---

### Author Response · Authors · 2024-11-27
**General Response 3:**

We thank all reviewers for their valuable feedback and suggestions. We have updated the PDF again with the following changes:

(1) Added results for batch size in Appendix D.3 and Table 10.

(2) Added results for training on CIFAR-100 from scratch in Appendix D.4 and Table 11.

(3) Added results regarding clip threshold in Appendix D.6 and Table 12.

(4) Updated Appendix C.1, C.2, and Table 13 for the experiments setup for training on CIFAR-100 from scratch.

(5) Updated Appendix F to incorporate discussions from the response.

---

### Meta-Review · Area_Chair_b9Sx · 2024-12-22

**Metareview:**

The authors propose an alternative to empirical MIA defences and DP-SGD and claims higher utility while making compromises in the theoretical guarantees in comparison to DP but in comparison to other works it still provides some. The authors provide both theoretical and empirical evidence that their method can be an alternative.

This paper is on the borderline. I personally like the work. However, there seems still a few issues that need to be addressed properly. For instance,  I agree with the reviewer 9rjs, in that because the owner's level of plausible deniability varies arbitrarily, this challenges the value of the approach in terms of operational meaning and interpretability, making this primarily an empirical defence. Providing a formal argument clearly describing the protection of individual samples will strengthen the paper for the next submission.

**Additional Comments On Reviewer Discussion:**

I agree with the reviewer 9rjs that the owner's level of plausible deniability varies arbitrarily, providing a formal argument clearly describing the protection of individual samples will be necessary.

---

### Decision · Program_Chairs · 2025-01-22

Reject